

# Analysis of 24 years of mesopause region OH rotational temperature
observations at Davis, Antarctica. Part 1: Long-term trends.
W. John R. French[1], Frank J. Mulligan[2] , and Andrew R. Klekociuk[1,3]
[1]Australian Antarctic Division, 203 Channel Hwy, Kingston, Tasmania, 7050, Australia
[2]Maynooth University, Maynooth, Co. Kildare, Ireland
[3]Department of Physics, University of Adelaide, Adelaide, 5005, Australia
*Correspondence to:* W. John R. French (john.french@aad.gov.au)





## Abstract

The long term trend, solar cycle response and residual variability in 24 years of hydroxyl nightglow rotational temperatures above Davis Research Station, Antarctica (68° S, 78° E) is reported. Hydroxyl rotational temperatures are a layer-weighted proxy for kinetic temperatures near 87 km altitude and have been used for many decades to monitor trends in the mesopause region in response to increasing greenhouse gas emissions. Routine observations of the OH(6-2) band P-branch emission lines using a scanning spectrometer at Davis station have been made continuously over each winter season since 1995. Significant outcomes of this most recent analysis update are (a) a record low winter-average temperature of 198.3 K is obtained for 2018 (1.7 K below previous low in 2009) (b) a long term cooling trend of 1.2 K/decade persists, coupled with a solar cycle response of 4.3 K/100 solar flux units and (c) we find evidence in the residual winter mean temperatures of an oscillation on a quasi-quadrennial (QQO) timescale which is investigated in detail in part 2 of this work.

Our observations and trend analyses are compared with satellite measurements from Aura/MLS version v4.2 level 2 data over the last 14 years and we find close agreement (a best fit) with the 0.00464 hPa pressure level values. The solar cycle response, long-term trend and underlying QQO residuals are consistent with the Davis observations. Consequently, we extend the Aura/MLS trend analysis to provide a global view of solar response and long term trend for southern and northern hemisphere winter season to compare with other observers and models.



## 1. Introduction

Long-term monitoring of basic atmospheric parameters is fundamentally important to understand natural, periodic and episodic variability in atmospheric processes, to provide data to verify increasingly sophisticated atmospheric models and to resolve and quantify perturbations due to global change on decadal to century timescales. Dynamical processes, including gravity waves, tides, planetary waves, large scale circulation patterns and quasi-periodic teleconnections (such as the quasi-biennial oscillation (QBO), El Niño Southern Oscillation (ENSO), and the Pacific Decadal Oscillation (PDO)), changes to the chemical composition and radiative balance (particularly due to anthropogenic emissions of greenhouse and chlorofluorocarbon gasses) and external forcing such as the 27-day solar rotation and 11-year solar activity cycle, all play significant roles (directly and through interactions) in defining and perturbing the mean state of the atmosphere. Decades of well calibrated measurements are required to accurately quantify variations and trends on these timescales.

Meteorological reanalyses derived from assimilation of a vast number of surface observations provide time-series for useful trend analyses for the lower atmosphere e.g. (Bengtsson et al., 2004). A few satellite based data sets are now also reaching multi-decadal timescales (e.g. the Thermosphere Ionosphere Mesosphere Energetics Dynamics satellite's Sounding of the Atmosphere using Broadband Emission Radiometry instrument (TIMED /SABER) (Mertens et al., 2003), and the Earth Observing System satellite Aura Microwave Limb Sounder (Aura/MLS) (Schwartz et al., 2008), that extend observations to the upper atmosphere. Of current and particular interest to climate science in the modern era are the atmospheric temperature trends in response to increasing global greenhouse gas emissions, principally from carbon dioxide ($CO_2$). Modelling studies over many years suggest that the sensitivity to $CO_2$ changes in the upper atmosphere, particularly at high latitudes, is



much larger than in the lower atmosphere (e.g. Roble (2000), the Canadian Middle
Atmosphere Model (CMAM) (Fomichev et al., 2007)) and the Hamburg Model of the
Neutral and Ionized Atmosphere (HAMMONIA) (Schmidt et al., 2006)).

Above the stratosphere, the low collision frequency means that $CO_2$ preferentially

radiates absorbed energy to space, resulting in a net cooling. Thus, the expected long-term
temperature trends in the mesosphere and lower thermosphere due to $CO_2$ are negative.
Ground based optical measurements of the Meinel emission bands of the hydroxyl (OH)
molecule produced by the exothermic hydrogen (H) – ozone ($O_3$) reaction (H + $O_3$ -> $OH^*$
+ 3.34 eV) have been used extensively over almost six decades as a method of measuring
atmospheric temperature in the vicinity of the mesopause (Kvifte, 1961; Sivjee, 1992; Beig
et al. 2003; Beig 2006; Beig et al. 2008; Beig 2011).  The emission is centred about 87 km
altitude and the rotational temperatures derived are representative of the kinetic
temperatures, weighted by the shape and width of the layer (~8 km full-width at half-
maximum (FWHM)). Temperatures thus obtained have always been considered ambiguous
to the extent that they are dependent on the altitude of the emitting layer, and they are
weighted by the altitude profile of that layer.  In the case of the OH* layer, different
vibrational bands are known to be weighted towards different altitude layers (von Savigny
et al. 2012), and on short time scales, individual bands vary in altitude with diurnal, semi-
diurnal, annual, semi-annual and solar cycle variations (García-Comas et al., 2017; Liu and
Shepherd, 2006; Mulligan et al., 2009).  Over long timescales (more than one solar cycle)
however, recent studies using satellite data (Gao et al., 2016; von Savigny, 2015) and OH
Chemistry-Dynamics (OHCD) models have shown that, the OH* layer altitude is
remarkably insensitive to changes in $CO_2$ concentration or solar cycle variation.  This
makes these measurements very valuable for monitoring long term changes in the
atmosphere.





This work provides an update on the solar cycle and long term trend analysis of the
OH rotational temperature measurements taken through each winter season at Davis
Research Station, Antarctica.  The dataset used here extends for 24 consecutive years and
this analysis includes a further 8 years of measurements since the previously published
trend assessment using these data ( French and Klekociuk, 2011). Here we expand on the
earlier analysis to provide a more detailed assessment of the solar response, trends and
variability in the Davis record in comparison with v4.2 measurements from the Microwave
Limb Sounder (MLS) on the Aura satellite (Aura/MLS) and a network of similar ground
based observations (coordinated by the Network for Detection of Mesospheric Change,
(NDMC) Reisin et al. 2014).
The outline of this paper is as follows. The instrumentation used and the acquired
rotational temperature data collection are presented in Sections 2 and 3. Analysis of solar
cycle response and the long-term linear trend is undertaken in Section 4 including
comparisons with other ground-based observers and satellite measurements. Discussion of
the results, summary and conclusions drawn are given in Sections 5 and 6, respectively.
We use the following terminology for the analysed temperature series in this manuscript.
From the measured temperatures and their nightly, monthly, seasonal or winter means,
*temperature anomalies* are produced by subtracting the climatological mean or monthly
mean (we fit solar cycle and linear trend to the anomalies), *residual temperatures*
additionally have the solar cycle component subtracted (used in discussion of long-term
trends) and *detrended temperatures* additionally have the long term linear trend subtracted
(used in discussion about remaining variability).



## 2. Instrumentation


A SPEX Industries Czerny-Turner grating spectrometer of 1.26 m focal length has
been used to autonomously scan the OH(6-2) P-branch emission spectra (λ839-851 nm) at
Davis (68.6° S, 78.0° E) each winter season over the last 24 years (1995-2018). Night-time
observations (sun > 8° below the horizon) are only possible between mid-February (~day
048) and end of October (~day 300) at Davis.
The spectrometer views the sky in the zenith with a 5.3° field-of-view and an
instrument resolution of ~0.16 nm, sufficient to separate $P_1$ and $P_2$ branch lines but not to
resolve their Lambda-doubling components. Observations are made regardless of cloud or
moon conditions and take of the order of 7 minutes to acquire a complete spectrum.
Spectral response calibration has been maintained by reference to several tungsten filament
Low Brightness Source units (a total of 4164 scans over the 24 years at Davis) which are
in turn cross referenced to national standard lamps at the Australian National Measurement
Institute (a total of 781 cross reference calibrations over 24 years). The response correction
accounts mainly for the fall-off in response of the cooled gallium arsenide (GaAs)
photomultiplier detector and amounts to 8.5% between the $P_1(2)$ and $P_1(5)$ of the OH(6-2)
band. The total change in spectral response correction over 24 years is less than 0.3%
(equates to less than 0.3 K for the $P_1(2) / P_1(5)$ ratio) despite changing the diffraction grating
in 2006 and four changes of the GaAs photomultiplier detector which are carefully
characterised over the years. The assigned annual calibration uncertainty is generally <0.3
K except for 1995 (1.8 K) due to calibration via a secondary calibration lamp and in 2002
(1.2 K) due to detector cooling problems. Further details of the instrument are contained in
Greet et al. (1997) and French et al. (2000).



## 3. Davis 24 year rotational temperature dataset

We use the three possible ratios from the $P_1(2)$, $P_1(4)$ and $P_1(5)$ emission line intensities to derive a weighted mean temperature. Intensity values are interpolated to a common time between consecutive spectra to reduce errors associated with the 7 minute acquisition cycle time. The weighting factor is the statistical counting error (based on the error in estimating each line intensity). $P_1(2)$ is corrected for the ~2% contribution by $Q_1(5)$. Line backgrounds are selected to balance the small auroral contribution of the N21PG and $N_2^+$ Meinel bands and solar Fraunhofer absorption for spectra acquired under moonlit conditions. Correction factors account for the difference in Lambda-doubling between the P-branch lines determined with knowledge of the instrument line shape from high-resolution scans of a frequency-stabilized laser.

Langhoff et al. (1986) transition probabilities are used to derive rotational temperatures (see French et al., 2000). Other published sets (e.g., Mies, 1974; Turnbull and Lowe, 1989; van der Loo and Groenenboom, 2007; Brooke et al., 2016) can change the absolute temperatures derived by up to 12 K, but does not significantly affect the trend analysis reported here. It should be noted however that comparison of absolute temperatures with other observations are significantly affected by different choices of transition probabilities.

Selection criteria limit extreme values of weighted standard deviation and counting error, slope and magnitude of the background and the rate of change of branch line intensities between consecutive scans. Further details of the rotational temperature analysis procedure are available in Burns et al. (2003) and French and Burns (2004).

Of over 624,000 measurements (typically ~26,000 profiles/year), 403,437 derived temperatures pass reasonably tight selection criteria (many low signal-to-noise ratio profiles taken through thick cloud or high background profiles around full moon are



rejected). These yield 5,309 nightly mean temperatures, where there are at least 10 valid
samples that contribute within ±12 hours of local midnight (~1850 Universal Time (UT)).
The time series spans two solar cycles (cycles 23 and 24) with peaks in 2001 and 2014.
Annual mean temperatures show a dependence on solar activity (see French and Klekociuk
(2011) for a comparison of different measures of solar activity with the Davis OH
temperature data). We use the 10.7 cm solar radio flux index (F10.7; 1 solar flux unit (sfu)
$= 10^{-22} \, W \, m^{-2} \, Hz^{-1}$) as our preferred measure of solar activity (F10.7 is fitted and subtracted
to examine residual variability). A plot of the nightly and winter mean temperatures with
the F10.7 time series used in this work is provided in Fig. 1.





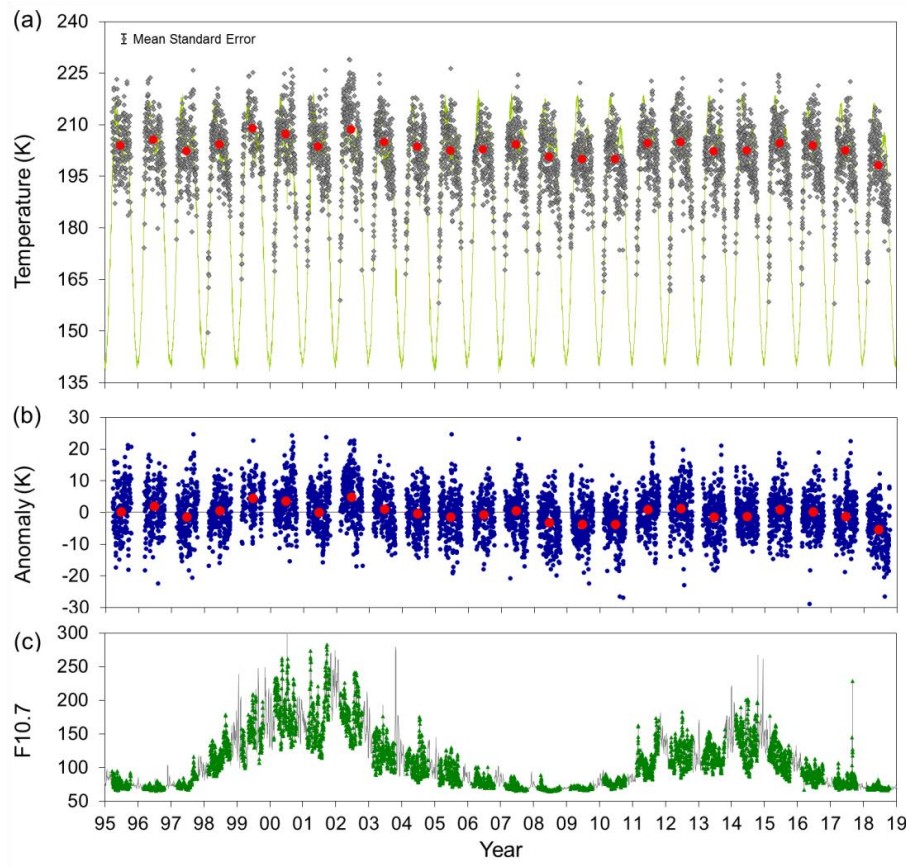


Figure 1 (a). Davis nightly mean temperatures (grey dots; 5309 samples) and winter
mean temperatures (D106-259; red points) plotted over the MSISE90 model temperature
for 68°S for seasonal reference (Hedin, 1991). (b). nightly mean and winter mean
temperature anomalies derived by subtracting the climatological mean (see text) and (c).
Daily mean F10.7 cm solar flux index (green points correspond to Davis OH temperature
samples)


A climatological mean is derived from a fit to the superposition of nightly mean
temperatures for all annual series. The climatological mean is characterised by a rapid
autumn transition (February-March) increasing at 1.2 K/day until a turn-over about 29
March (day of year D088), a slow winter decline (April-September) of -0.4 K/day that is
punctuated by mid-April (~D113) and mid-August (~D227) dips corresponding to
reversals in the mean meridional flow (Murphy et al., 2007), followed by a rapid spring
transition (October-November) of -1.0 K/day. Figure 2 shows the superposed nightly


means for each year and the climatological mean fit. Subtracting the climatological mean
produces 5309 nightly mean temperature anomalies. Winter mean temperatures are
calculated over the interval from 15 April (D106) to 15 September (D259) which avoids
the winter to summer transition intervals and lower numbers of nightly observations due to
the shorter night length.

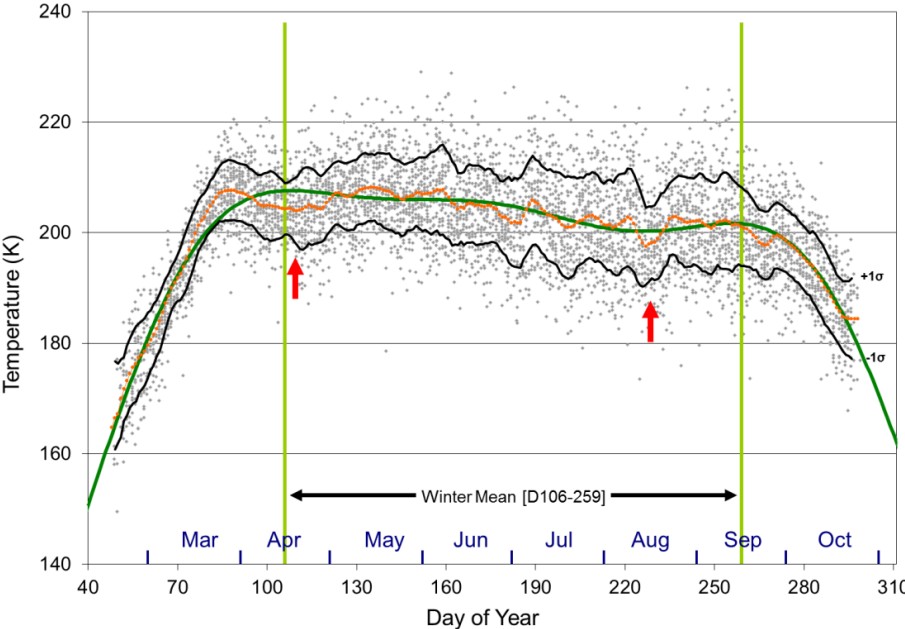


Figure 2. Superposed nightly mean temperatures from 1995 to 2018 [gray points]
and a 5-day running mean which represents the climatological mean [orange line] with 1σ
intervals [black lines]. The seasonal variation [green annual, semi-annual, ter-annual fit] is
characterised by a rapid autumn transition (Feb-Mar) increasing at 1.2 K/day until a turn-
over about 29th-March (day 088), a slow winter decline (Apr-Sep) of -0.4 K/day,
punctuated by mid-April and mid-August dips [indicated by red arrows], followed by a
rapid spring transition (Oct-Nov) of -1.0 K/day. Green vertical lines mark the calculation
region for winter mean temperatures (outside the winter to summer transition intervals).


## 4. Trend Assessment

4.1 Davis winter mean trends

Winter mean temperature anomalies over the 24 years of observations are plotted
in Fig. 3a. The time series is fitted with a linear model containing a solar cycle term (F10.7)
and long term linear trend. This model yields a solar cycle response coefficient of 4.30 ±
1.02 K/100sfu (95% confidence limits 2.2 K/100sfu < S < 6.4 K/100sfu) and a long term
linear trend of -1.20 ± 0.51 K/decade (95% confidence limits -0.14 K/decade < L < -2.26
K/decade) and accounts for 58% of the temperature variability.

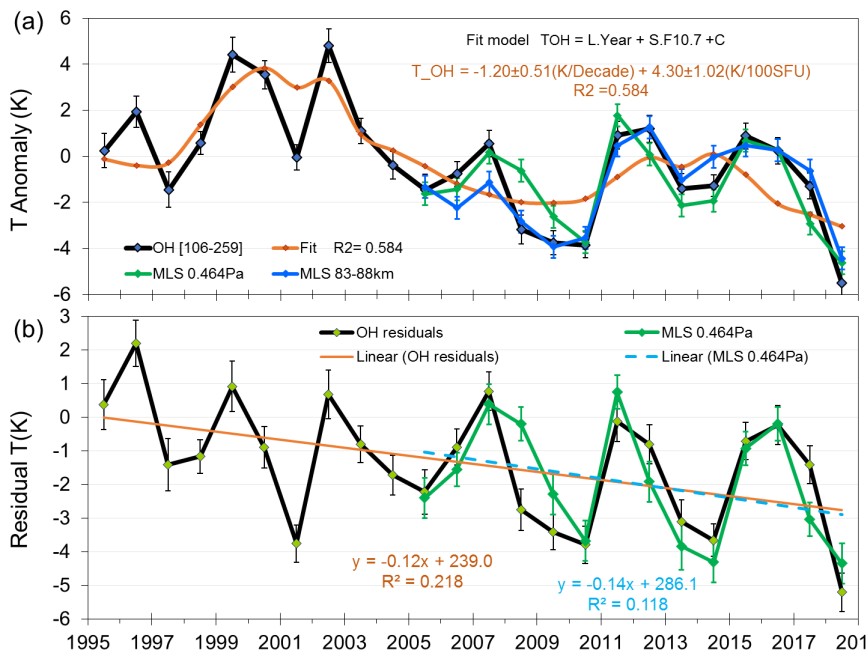

Figure 3 (a). Winter mean (D106-259) temperature anomalies (black line) for Davis
station (68°S, 78°E) fitted with a linear model containing a solar cycle term (F10.7cm flux)
and long term linear trend (orange line). Fit coefficients are 4.30±1.02 K/100sfu (95%
confidence limits 2.18 to 6.42 K/100sfu) and -1.20±0.51 K/decade (95% confidence limits
-0.14 to -2.26 K/decade) respectively and account for 58% of the temperature variability.
Also plotted (from 2005) are Aura/MLS temperature anomalies derived from the AMJJAS
means of all satellite observations within 500 km of Davis station. (b) As for (a), but with
the solar cycle component removed to better reveal the long term trend and quasi-
quadrennial oscillation (QQO). OH residuals (black line) are compared with Aura/MLS
temperature residuals at the 0.0046 hPa level, corrected with the same solar cycle
component as used for the Davis OH measurements.



We note that a new record low winter-mean temperature was set for the Davis
measurements in 2018, with a value of 198.3 K, which is 1.7 K below the previous
minimum recorded in 2009 (200.0 K). This is not entirely due to the low solar activity in
2018 (winter mean flux of 70.4 sfu) as both 2008 (66.9 sfu) and 2009 (69.1 sfu) had lower
mean flux and comparable years 1996 (70.6 sfu) was 7.4 K warmer (205.7 K) and 2007
(71.9 sfu) was 6.1 K warmer (204.4 K).
Extracting the solar cycle contribution from the time series yields the long term
linear trend and residual variability plotted in Fig. 3b. It is apparent from this plot that a
significant oscillation on an approximately 4-year (quasi-quadrennial) timescale remains.
A least-squares fit of a sinusoidal function to the data yields a period of 4.2 years and peak-
peak amplitude of ~3 K.  This feature will be examined in detail in Part 2 of this work
(French et al., 2019).
Distributions of the nightly mean residual temperatures for each year are shown for
comparison in Fig. 4. Histogram colour scale indicates the winter mean temperature from
warmest year (1999; red) to coldest year (2018; blue). Distributions vary between years
from sharp normal distributions (e.g., 1998, 2007, 2016), to broad flat distributions (e.g.,
1996, 1997), to skewed or double peaked distributions (e.g., 2004, 2012, 2014, 2018).
These differences can be attributed to the variability in large scale planetary wave activity
from year to year (French and Klekociuk, 2011)

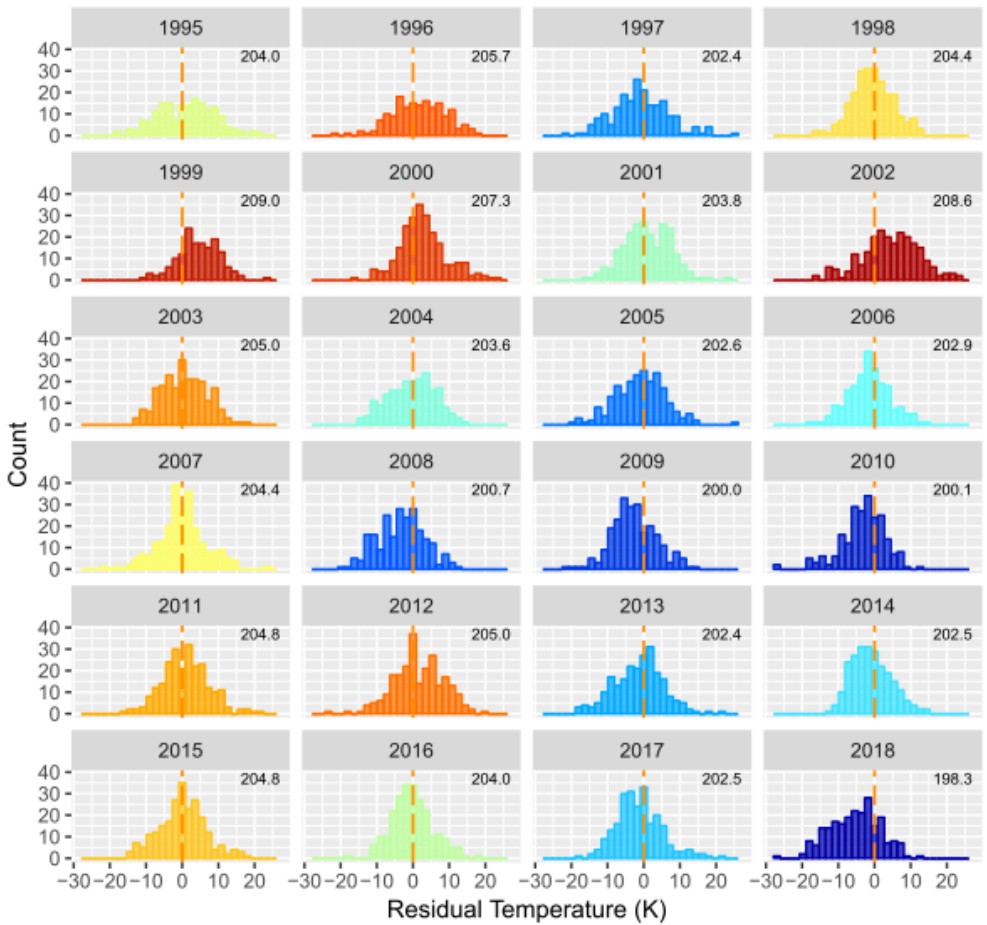


Figure 4. Histograms of nightly mean residual temperatures showing the distribution about the mean winter temperature (annotated in top right corner) coloured from red (warmest year: 1999) to blue (coldest year: 2018).

4.2 Seasonal variability in trends.

Seasonal trend coefficients are also somewhat variable. Figure 5 shows the seasonal variability in solar cycle and long-term trend coefficients derived using a 60 day sliding window, and as monthly trends, compared to the winter mean trends (red lines) derived for Fig. 3. Seasonal solar response shows a maximum in May-June (~5 K/100sfu) and minimum around August (~2 K/100sfu). Note that April and August temperatures are affected by the characteristic dips seen in the climatological mean during these months (see

Fig. 2). Linear trend coefficients show maximum cooling responses in April-May (~ -1.3
K/decade) and in August-October (~ -2.5 K/decade). Virtually no long-term cooling trend
is apparent for the midwinter months of June-July.

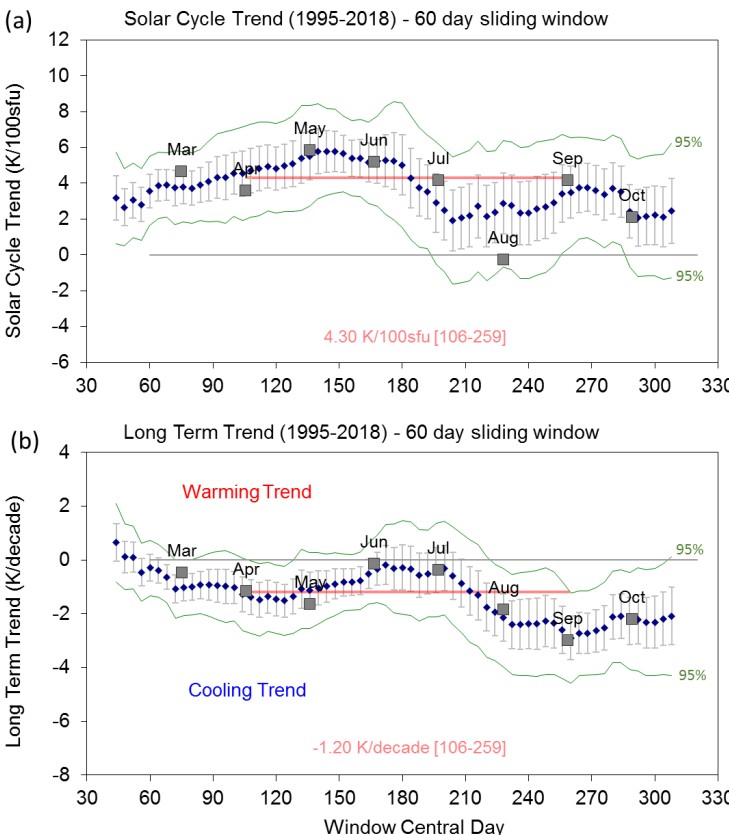


Figure 5. The seasonal variability in (a) solar cycle and (b) long-term trend
coefficients derived using a 60 day sliding window (blue dots), and as monthly trends (grey
boxes) compared to the winter mean trends (red lines) derived for Fig. 3. The green lines
show the confidence limits (95%) for the trend coefficients.

4.3 Aura/MLS trend comparison

For comparison with the Davis trend measurements, we use version v4.2 level 2

data from the Microwave Limb Sounder (MLS) instrument on the Earth Observing System
Aura satellite launched in July 2004 (Schwartz et al., 2008). Aura/MLS provides almost
complete global coverage (82° S-82° N) of limb scanned vertical profiles (~5-100 km) of



temperature and geopotential height derived from the thermal microwave emissions near
the spectral lines 118 GHz $O_2$ and 234 GHz $O^{18}O$. Previous comparisons of these data
with MLS v2.2 temperatures were conducted by French and Mulligan, 2010.

Over-plotted in Fig. 3a (extending from 2005) are the equivalent Aura/MLS mean

temperature anomalies computed by averaging all observations within 500 km of Davis,
for months April to September (AMJJAS) over altitudes 83-88 km (blue line, obtained
from a linear interpolation of Aura/MLS geopotential height profiles to geometric height
in 1 km steps) and at the 0.00464 hPa (native Aura/MLS retrieval) pressure level (green
line). The Aura/MLS data were selected according to the quality control recommendations
described in Livesey et al. (2018). Approximately 60 samples/month are coincident within
this range. We see close agreement considering that at these altitudes the vertical resolution
(FWHM of the averaging kernel) of Aura/MLS in approximately 15 km (Schwartz et al.,
2008). The Aura/MLS measurements closely follow the solar response, the magnitude and
period of the quasi-quadrennial oscillation (QQO) and the underlying long-term linear
trend. Statistically, the closest agreement is with the 0.00464 hPa pressure level and this is
over-plotted on Fig. 3b corrected for the solar cycle response determined from the Davis
OH measurements. The linear long-term trend fit for Aura/MLS over 14 years is -1.43
K/decade (comparable to the -1.2 K/decade for the 24 years of Davis OH measurements)
but clearly the underlying QQO variability has a significant effect on the fit.

It is important to note that the winter mean residual trend coefficients in Fig. 3b are

derived as a mean across 6-months of significantly varying solar and long-term responses.
Nevertheless, the residual QQO signature remains readily apparent in the 60-day sliding
window means through April to July [AMJJ] although somewhat breaking down in August
to October [ASO].





We examine the QQO feature in more detail in the second part of this work (French,
et al., 2019), but here, given the close agreement of Davis and Aura/MLS trends in Fig. 3b,
we apply the same model fit procedure to derive Aura/MLS solar cycle and linear long-
term trend coefficients to obtain a global picture of trends at the hydroxyl layer equivalent
pressure level (0.0046 hPa). Figure 6 shows global trends determined by averaging
Aura/MLS pressure level 0.0046 hPa temperature anomalies into a 5° x 10° (latitude x
longitude) grid, over Southern Hemisphere (SH) winter months (AMJJAS; top panels)
compared to Northern Hemisphere (NH) winter months (October-March; ONDJFM;
bottom panels).  Each grid box has been corrected for the solar cycle response determined
from a linear regression of temperature to F10.7 over the 14 years of Aura/MLS
measurements. The long-term linear trend (left-hand panels) and solar cycle response
(right-hand panels), for each grid box, together with their corresponding zonal means are
presented. The maps contain some interesting features; enhanced bands of solar activity
response occur at mid-latitudes in both winter hemispheres although strongest in the SH
(colour scales are the same for each hemisphere). Minima in sensitivity to solar forcing
occur over the equator and the poles. Long-term trends over the Aura/MLS era are not
globally uniform. While the global mean trend for the SH winter [AMJJAS] is -0.31
K/decade, there are regions of warming, notably around the equator, southern Africa,
Europe and the Atlantic ocean and strongest cooling over Antarctica and northern Canada.
For the NH winter [ONDJFM] the global mean is -0.11 K/decade with generally global
cooling, except for warming over Antarctica, Europe, southern Africa and the northern
Pacific Ocean.

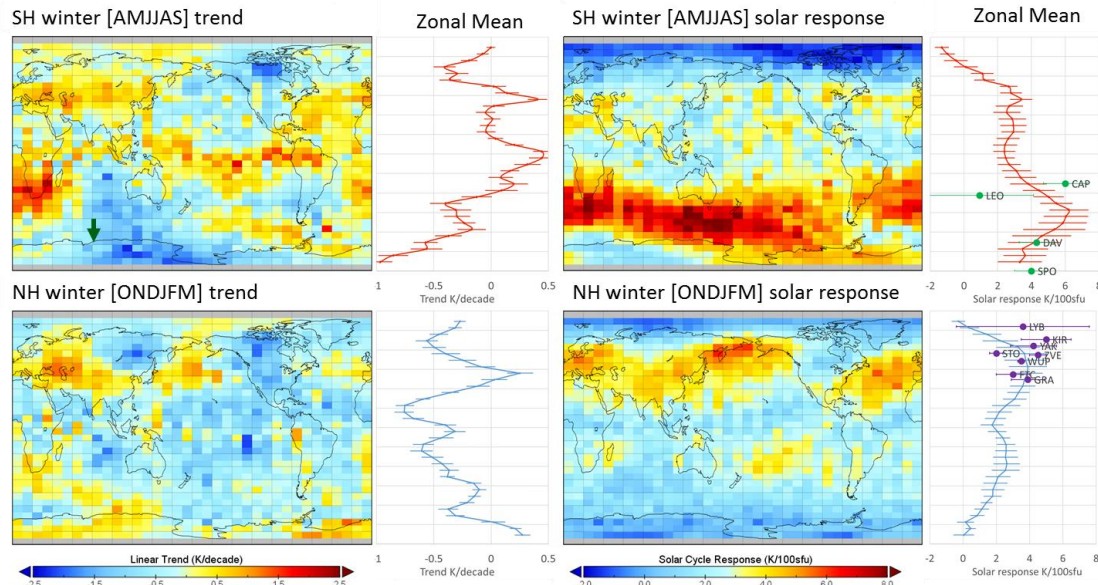


Figure 6. Global temperature trends and solar cycle responses, together with their corresponding zonal means determined from 14 years of MLS v4.2 pressure level 0.0046hPa (hydroxyl layer equivalent), averaged into 5° latitude x 10° longitude grid, and over southern hemisphere winter months (AMJJAS; top panels) compared to northern hemisphere winter months (ONDJFM; bottom panels). The linear trend and solar cycle response coefficients have been derived individually for each grid box from Aura/MLS over 14 years with no lag. The green arrow in panel 1 indicates the Aura/MLS comparison with Davis shown in Fig 1B. Solar response coefficients from other observers are indicated on the zonal solar response plots (see text for site information)

4.4 Trend comparisons with other ground based observations

It is useful to compare these Aura/MLS derived solar response and trend coefficients with other observations, carefully bearing in mind that these observations may span different time intervals than available in the Aura/MLS measurement epoch. At Davis the solar cycle response (indicated by the green dot DAV in Fig. 6) determined over 24 years matches well with the zonal mean at 68° S determined from the Aura/MLS measurements. Davis appears to be on the poleward boundary of the strong band of solar



sensitivity (~40-70° S) in the SH winter.  The long-term trend at Davis is marked by the
green arrow on the left-hand upper panel in Fig 6, and as we have seen from Fig. 3, agrees
well with Aura/MLS.

Table 1 summarises the data-span, derived long term trend, and solar cycle

coefficients from a collection of ground-based observers. Where new results are available
these have been updated from Table 2 in French and Klekociuk (2011) and as compiled in
Beig et al. (2008). The majority of these observations agree well with the Aura/MLS solar
trend evaluated here, given it is a zonal mean response.

| Site | Data Span | Trend K/decade | Solar response K/100sfu | Reference |
|---|---|---|---|---|
| Longyearbyen (LYB, 78°N, 16°E) | 1983-2013 | -0.2±0.5 | 3.6±4.0 | Holmen et al. (2014) |
| Kiruna (KIR, 68°N, 21°E) | 2003-2014 | -2.6±1.5 | 5.0 ± 1.5 | Kim et al. (2017) |
| Yakutia (YAK, 63°N, 129°E) | 1999-2013 | Not Significant | 4.24±1.39 | Ammosov et al. (2014) |
| Stockholm (STO, 57°N, 12°E) | 1991, 1993-1998 | Not Determined | 2.0±0.4 | Espy et al. (2011) |
| Zvenigorod (ZVE, 56°N, 37°E) | 2000-2016 | -0.07±0.03 | 4.5±0.5 | Perminov et al. (2018) |
| Wuppertal (WUP, 51°N, 7°E) | 1988-2015 | -0.89±0.55 | 3.5±0.21 | Kalicinsky et al. (2016) |
| Fort Collins (FTC, 41°N, 105°W) | 1990-2018 | -2.3±0.5 | 3.0±1.0 | Yuan et al. (2019 in press) |
| Granada (GRA, 37°N,3°W) | 2002-2015 | -0.6±2.0 | 3.9±0.1 | Garcia-Comas et al. (2017) |
| Cachoeira Paulista (CAP, 23°S, 45°W) | 1987-2000 | -1.08±0.15 | 6±1.3 | Clemesha at al. (2005) |
| El Leoncito (LEO, 32°S, 69°W) | 1998-2002 | Not Determined | 0.92±3.2 | Scheer et al. (2005) |
| Davis (DAV, 68°S,78°E) | 1995-2018 | -1.20±0.51 | 4.30±1.02 | This Work |
| South Pole (SPO, 90°S) | 1994-2004 | 0.1±0.2 | 4.0±1.0 | Azeem et al. (2007) |

Table 1. A comparison of solar cycle response and temperature trend observations from
the ground-based OH observer network with updates since 2011 where available.

As some observers have found, there is a significant question about a time delay in

the OH layer temperature response to solar forcing via the various solar absorption
mechanisms in the atmosphere. The major absorbers and altitude of solar extreme
ultraviolet radiation are molecular oxygen (Schumann-Runge continuum, 80-130 km,



Schumann-Runge electronic and vibrational bands, 40-95 km, Herzberg continuum, below
50 km) and ozone (Hartley-Huggins bands, below 50 km).
We have previously found a lag of around 160 days (F10.7 leads temperature) is
best fit to the linear model (French and Klekociuk, 2011), others find shorter: 80 days at
Longyearbyen, Svalbard (Holmen et al. 2014), or larger lags: 25 months at Maimaga station,
Yakutia (Ammosov et al. 2014; Reisin et al. 2014). Recalculating the long term trends for
Aura/MLS assuming a uniform global solar response (as for Davis), or with a 160 day lag
and zonal mean solar response (see supplementary material) does not significantly change
the warming and cooling patterns shown in Fig. 2, but the lag does reduce the cooling trend
(on average by 0.16 K/decade for the southern hemisphere (SH) winter and 0.11 K/decade
for the northern hemisphere (NH) winter) and increases the fit error.
Beig (2011a, 2011b) in their reviews of long-term trends in the temperature of the
mesosphere and lower thermosphere (MLT), highlight the difficulty of distinguishing
between the anthropogenic and solar cycle influences. In their results, mesopause region
temperature trends were found to be either slightly negative or zero. At that time, it was
believed that the solar response becomes stronger with increasing latitude in the
mesosphere with typical values in the range of a few degrees per 100 solar flux units in the
lower part of the mesosphere but reaching 4-5 K/100 sfu near the mesopause. More recent
studies using longer data sets (Ammosov et al. 2014; Holmen et al. 2014; Perminov et al.
2018) and satellite data (Tang et al. 2016) have reinforced that view.
Trend breaks began to appear in mesopause region temperatures in 2006
(Offermann et al. 2006), and these continue until now in certain locations (e.g., Jacobi et
al., 2015; Kalicinsky et al., 2018; Yuan et al., 2019). These can be quite varied from site
to site, ranging from -10 K/decade to +5 K/decade. Some of these estimates simply suffer
from lack of observations (measurement spans less than a solar cycle). Few are longer than



2 solar cycles, but those of note are included in Table 1.  OH temperature trend studies in
the southern hemisphere are less common.  Reid et al. (2017) report MLT-region nightglow
intensities, temperatures and emission heights near Adelaide (35° S, 138° E), Australia.
Five years (2001-2006) of spectrometer measurements using OH(6-2) and $O_2$(0-1)
temperature are compared with 2 years of Aura/MLS data and 4.5 years of SABER data.
Venturini et al. (2018) report mesopause region temperature variability and its trend in
southern Brazil (Santa Maria, 30° S, 54° W), based on SABER data over the period 2003-
2014.  Nath and Sridharan (2014) examined the response of the middle atmosphere
temperature to variations in solar cycle, QBO and ENSO in the altitude range 20-100 km
and 10-15° N latitude using monthly averaged zonal mean SABER observations for the
years 2002-2012. They found cooling trends in most of the stratosphere and the mesosphere
(40–90 km).  In the mesosphere, they found the temperature response to the solar cycle to
be increasingly positive above 40 km.  The temperature response to ENSO was found to
be negative in the middle stratosphere and positive in the lower and upper stratosphere,
whereas it appeared largely negative in the height range 60–80 km and positive above 80
km.



# 5. Discussion

5.1 Relationship between Davis trends and $CO_2$ change.

Our updated trend assessment over 24 years yields a cooling rate of -1.20±0.51 K/decade for the mean winter [D106-259] temperatures in the hydroxyl layer above Davis. A slightly greater rate of -1.32±0.45 K/decade is derived if the full year [D040-310] of observations are included in the annual means. Over the same period, annual mean surface $CO_2$ volume mixing ratios (VMRs) increased from 360.82 ppm [1995] to 408.52 ppm [2018] (Mauna Loa values from Global Greenhouse Gas Reference Network www.esrl.noaa.gov/gmd/ccgg/trends/), an increase of 47.7 ppm or 13.2% (19.9 ppm per decade or 5.5% per decade). Qian et al. (2019) quote a $CO_2$ trend figure of 5.2%/decade (or 5.1 % if the seasonal variation is removed before the linear trend calculated) based on measurements made by TIMED/SABER from 2002-2015. If the primary factor for the observed temperature trend is considered to be $CO_2$ radiative cooling, a coefficient of -0.06 K/ppm$CO_2$ or -0.22 K/%$CO_2$ is implied. This is approximately twice the value obtained by (Huang, 2018) (her Figure 2) who employed a linear scaling of the result of a doubling of $CO_2$ concentration by (Roble and Dickinson, 1989). A $CO_2$ increase of 26.5% from 1960 to 2015 was accompanied by a temperature decrease of 1.4% at an altitude of 89.4 km near Salt Lake city, Utah (18° N, 290° E).

$CO_2$ is well mixed through the lower atmosphere with a constant VMR up to about 80 km. Above this height, diffusion and photolysis processes begin to have an effect, reducing the VMR (Garcia et al., 2014) but these processes vary with latitude and season (Rezac et al. 2015; López-Puertas et al., 2017).

Several studies of $CO_2$ VMR using profiles from the Atmosphere Chemistry Experiment Fourier Transform Spectrometer (ACE-FTS) and Sounding of the Atmosphere using Broadband Emission Radiometry (SABER) satellite instruments, reported



considerably larger rates of change of $CO_2$ in the upper atmosphere, increasing from about
5% per decade at 80 km to 12% per decade at 110 km (Emmert et al., 2012; Garcia et al.,
2016; Yue et al., 2015). However, more recent analysis of the ACE-FTS and SABER $CO_2$
data with different deseasonalizing procedures have shown an average rate of 5.5% per
decade in the 80-110 km region, consistent with surface rates (Qian et al., 2019; Rezac et
al., 2018).

In a recent summary of progress in trends in the upper atmosphere, Laštovička

(2017) identified greenhouse gases, particularly $CO_2$ as the primary driver of long-term
trends there.  The important secondary trend drivers in the mesosphere and lower
thermosphere (MLT) are stratospheric ozone, water vapour concentration and
atmospheric dynamics.  The overall effect of greenhouse gases at mesospheric altitudes is
radiative cooling.  Temperature trends are predominantly negative, and recent progress in
understanding the magnitude of the cooling have arisen from confirmation and
quantification of the role of ozone.  In the mesopause region, about two thirds of the
cooling is attributed to increases in $CO_2$ concentration and one third to changing
concentration of ozone in the stratosphere (Lübken et al., 2013).  Increases in water
vapour concentration are considered a secondary but non-negligible effect particularly in
the lower thermosphere (Akmaev et al. 2006).  Trends in ozone vary as a function of both
altitude and latitude, with positive trends dominating in the lower stratosphere and
mesosphere.

Huang (2018) examined the influence of $CO_2$ increase, solar cycle variation and

geomagnetic activity on airglow from 1960 to 2015 using two airglow chemistry dynamics
models (OHCD – OH chemistry dynamics, and MACD – multiple airglow chemistry
dynamics).  As expected, the results showed that airglow intensity and peak volume
emission rate (VER) are in phase and have a linear relationship with F10.7 values, whereas





$CO_2$ increase leads to a slowly decreasing trend in OH(8-3) airglow intensity. OH(8-3)
peak altitudes of the VER are unaffected by increases in $CO_2$ concentration, and are only
slightly affected by the F10.7 cycle, with slightly lower peak altitudes when F10.7 is <100
SFU. Surprisingly, OH VER peak heights showed a significant inverse relationship with
geomagnetic activity as measured by the Ap index. We find no significant correlation of
the $T$-residual from Davis with the Ap index for the months of AMJJAS.

Lübken et al. (2013) present the results of trend studies in the mesosphere in the

period 1961-2009 from the Leibniz-Institute Middle Atmosphere (LIMA) chemistry-
transport model which is driven with European Centre for Medium–Range Weather
Forecasts (ECMWF) reanalysis below 40 km, and observed variations of $CO_2$ and $O_3$.
They find that $CO_2$ is the main driver of temperature change in the mesosphere, with $O_3$
contributing approximately one third to the trend. Linear temperature trends were found
to vary substantially depending on the time period chosen primarily due to the influence of
the complicated temporal variation of ozone. The trend effect of dynamics was found to
be very slightly negative in the mesosphere, but very small compared with the radiatively
induced trends. At the mesopause, the trend due to dynamics was positive and significantly
larger (~1 K/decade). These results were found to be in good agreement with observations
from lidars, Stratospheric Sounding Units (SSU) (Randall et al., 2009) and radio reflection
heights which have decreased by more than 1 km in the last 50 years due to shrinking in
the stratosphere/lower mesosphere caused by cooling. Figure 3 of (Lübken et al., 2013)
show a monotonically increasing trend on $CO_2$ compared with a much more complicated
temporal ozone variation (essentially constant until 1980, a rapid decrease from 1980-1995,
followed by an increase since then.

A recent paper by Hervig et al. (2019) report on the absence of a solar signal

correlated response in polar mesospheric clouds (PMCs) in the summer mesopause



following 2002. PMCs are controlled by temperature and water vapour. At solar maximum,
temperatures are expected to be higher and water vapour lower, thereby leading to less
PMCs at solar maximum. This anti-correlation was evident in satellite data until 2002, but
has been absent since then. The main cause for the diminished solar cycle in PMCs at 68°
N and 68° S appears to be the dramatic suppression of the solar cycle response in water
vapour. The solar cycle response of temperature also decreases after 2002, but has a much
lower effect on PMCs than the water vapour.

The Whole Atmosphere Community Climate Model (WACCM) extended into

thermosphere (upper boundary ~700 km) (WACCM-X) was used by Qian et al. (2019)
(with the lower atmosphere constrained by reanalysis data) to investigate temperature
trends and the effect of solar irradiance on temperature trends on the mesosphere during
the period 1980-2014. The overall temperature trend in the mesopause region at 85 km
was statistically insignificant at -0.46 ± 0.60 K/decade. Solar irradiance effects on the
global average temperature are positive and decrease monotonically with decreasing
altitude from a value of ~3 K/100 sfu in the lower thermosphere to ~1 K/100 SFU at 55
km. This is readily explained by the decreasing external energy from the Sun with reducing
altitude. A monthly mean global average trend of 2.46 K/100 sfu is quoted for the
mesopause near 85 km. The mesosphere is affected by solar irradiance directly from local
heating through absorption of radiation, and indirectly through dynamics by its effects on
the geostrophic winds which control the upward propagation of gravity waves and
planetary waves generated in the troposphere. Zonal mean temperatures show significant
variability as a function of altitude, latitude and season. Qian et al. (2019) provide globally
averaged temperature trend values as a function of altitude and latitude for each month
some of which are statistically significant. Solar cycle effects on temperature are in
reasonable agreement with the OH(6-2) temperatures shown in Figure 5 with positive


values ranging from ~3-5 K/100 sfu, the largest values occurring in July and October.  The
long-term trend is predominantly negative with values in the range -1 to -3 K/decade with
the largest cooling occurring in March and September at the latitude and altitude of the OH
temperatures measured at Davis Station.  WACCM-X shows slightly positive trend values
in the months of February, November and December at Davis Station, but OH(6-2)
temperature data are not available in these months. The September maximum in cooling is
in reasonable agreement with the Davis measurements shown in Figure 5 of this work.

More recent results from Garcia et al. (2019) using WACCMv4 free-running

(coupled ocean) simulations for the period 1955-2100 using IPCC RCP 6.0 attribute the
changes in the trends of the temperature profile to monotonic increases in $CO_2$
concentration together with a decrease in $O_3$ until 1995 followed by subsequent increase.
Garcia et al. (2019) assign half of the stratopause negative temperature trend to ozone
depleting substances.  At the mesopause, the global mean trend in temperature is
approximately -0.6 K/decade.  Solar cycle signals at the mesopause are in the range 2-3
K/100 sfu with slightly higher values in the southern polar cap.  Very large seasonal trends
in temperature at all altitudes are associated with the development of the Antarctic ozone
hole.  Trends are largest in the November-December period, and teleconnections are made
with the upper mesosphere via GW filtering by the zonal wind anomaly in the southern
polar cap

5.2     Trend breaks.

When analysing long-term trends, several authors (Lübken et al., 2013; Qian et al.,

2019) emphasise the importance of specifying the length of the time period, as well as the
beginning and end of the period, because trend drivers can be different for different periods
(e.g., Yuan et al., 2019).  Yuan et al. (2019) report long-term trends of the nocturnal





mesopause temperature and altitude from LIDAR observations at mid-latitude (41-42° N,
105-112° W) in the period 1990-2018. They divided their observations into two categories,
the high mesopause (HM) above 97 km during the non-summer months, mainly formed by
radiative cooling, and the low mesopause (LM) below 92 km during the non-winter months
generated by mostly by adiabatic cooling. This idea of the mesopause at two different
altitudes is well established (e.g., von Zahn et al., 1996; Xu et al., 2007; Thulasiraman and
Nee, 2002). Although Yuan et al. (2019) obtained a cooling trend of more than 2K/decade
in the mesopause temperature along with a decreasing trend in mesopause height since
1990, the temperature trend is statistically insignificant since 2000.

Trend breaks have been reported at other mid-latitude stations (Offermann et al.,

2006) where a discontinuity was found in the overall trend in the year 2001/2002. Using
some of the same data as Offermann et al. (2006), Kalicinsky et al. (2016) reported a trend
break in the middle of 2008. Before the break point, there is a clear negative trend reported
to be -2.4 ± 0.7 K/decade, whereas after 2008, a large positive trend of 6.4 ± 3.3 K/decade
is deciphered. Two possible explanations are suggested for the trend break: the first is that
it is the result of a combination of the solar cycle and a long period oscillation such as the
22-year Hale cycle of the Sun. A second possible explanation of the very substantial
change in the trend at 2008 is a combination of the solar flux with a sensitivity of 4.1 ± 0.8
K /100 SFU together with a long period oscillation 24-26 years with an amplitude of about
2K. Kalicinsky et al. (2018) find support for this idea in the identification of a quasi-
decadal oscillation in the summer mesopause over Western Europe in plasma scale height
observations (near 80 km altitude) which are in anti-correlation with the potential
oscillation in temperature from OH* measurements. The anti-correlation in the two data
sets is explained on the basis of the fact that they originate below (plasma scale height data)
and above (OH* temperature data) the temperature minimum in the mesopause region in



summer. Jacobi et al. (2015) find that the long-term behavior of both meridional and zonal
winds at 90-95 km in northern mid-latitude stations exhibit trend breaks in summer near
1999, although the winter data are well described by a single linear trend over the years
1980- 2015. We find no obvious sign of a discontinuity in the trend obtained in the Davis
data.

5.3    Effect of changes in the OH*-layer height

There is widespread acceptance that cooling of the middle atmosphere due to

increases in $CO_2$ concentration has resulted in shrinking of the middle atmosphere (e.g.,
(Grygalashvyly et al., 2014; Sonnemann et al., 2015). This does raise the question
however of whether the OH* layer is fixed to a constant pressure level rather than a
constant altitude. There are mixed reports on this topic. In a long-term study of the
effects of chemistry, greenhouse gases, and the solar modulation on OH* layer trends
using the Leibniz Institute Middle Atmosphere (LIMA) chemistry-transport model
covering the period 1969 to 2009, Grygalashvyly et al. (2014) reported a downward shift
in the OH*-layer by about 0.3 km/decade in all seasons due to shrinking of the middle
atmosphere resulting from radiative cooling by increasing $CO_2$ concentrations. Wüst et
al. (2017) report a descent in the mean altitude of the OH* layer of 0.02 km/ year from 14
years of SABER data (2002-2015) in the alpine region of southern Europe (44–48° N, 6–
12° E). They refer to a paper by Bremer and Peters (2008) which reports low frequency
reflection heights (ca. 80-83 km) between 1959 and 2006 and derive a figure of 0.032
km/year.

Sivakandan et al. (2016) have published a long-term variation paper on OH peak

emission altitude and volume emission rate over Indian low latitudes using SABER data.



A weak decreasing trend of 19.56 m/year was reported for the peak emission altitude of
the night-time OH*-layer.
A vertical shift of the OH* layer either upward or downward gives rise to a change

in the emission weighted temperature which is measured by ground-based optical
instruments (French and Mulligan, 2010; von Savigny, 2015). Von Savigny (2015)
reported no apparent trend or solar cycle in OH emission altitude at the local time of the
SCIAMACHY nighttime observations in the period 2003-2011. However, Teiser and von
Savigny (2017) found evidence of an 11-year solar cycle in the vertically integrated
emission rate and in the centroid emission altitude of both the OH(3-1) and OH(6-2) bands
in SCIAMACHY data. Gao et al. (2016) found no evidence that the OH* peak heights are
affected by solar cycle in 13 years of TIMED/SABER data, and deduced that the solar
cycle variation of temperature obtained from ground-based OH nightglow observations
were essentially immune from the OH emission altitude variations. Huang (2018) found
no systematic response of airglow $O(^1S)$ green line, $O_2(0,1)$, or OH(8-3)) VER peak heights
with the F10.7 solar cycle using two airglow models OHCD and MACD-90. The Huang
(2018) result is supported by Gao et al. (2016) using TIMED/SABER data and by von
Savigny (2015) using SCIAMACHY data. These confirmations of the remarkable long-
term stability of the peak altitude of the OH*-layer in an atmosphere with increasing $CO_2$
concentration and changing solar radiation are essential for the use of long-term studies of
mesopause region temperatures derived from ground-based OH* optical measurements.

We have examined the altitude of the OH* layer during the period 2002-2018

using the OH-B channel volume mission rate (VER) from TIMED/SABER (version 2.0)
sensitive in the wavelength range 1.56-1.72 μm, which includes mostly the OH(4-2) and
OH(5-3) bands. All VER altitude profiles between day 105 and day 259 that satisfied the
selection criteria (tangent point within 500 km of Davis and solar zenith angle > 97°),





employed by French and Mulligan (2010) were used to determine the altitude of the
layer. The altitude of the peak was obtained from a Gaussian profile fitted to the VER
profile (for more details, see French and Mulligan, 2010). The slope of the best fit line to
the winter annual average peak altitude was -0.02 ± 0.02 km/ year as shown in Figure 7,
i.e., no significant change in altitude of the layer over the period in agreement with the
result of Gao et al. (2016).

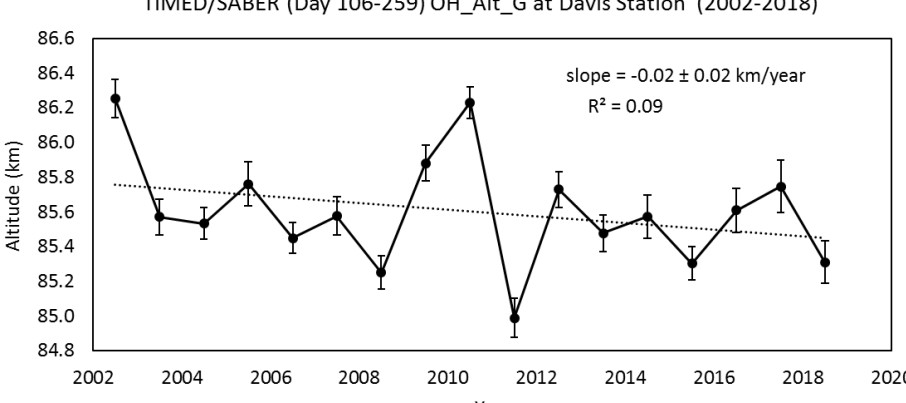


Figure 7. The trend in the mean winter OH layer altitude, derived from TIMED/SABER
(version 2.0) OH-B channel volume emission rate. The slope of the best fit line is -0.02 ±
0.02 km/year, i.e., no significant change in altitude of the layer over this interval.

5.4     Global solar cycle and linear trends

The trend measured at Davis is well matched with the result from Aura/MLS over

14 years for the southern hemisphere (SH) winter months (AMJJAS) at the 0.0046 hPa
level. Clearly though, applying the same analysis to the global temperature field reveals
that trends are not globally uniform (Fig 6). In the SH winter the most significant cooling
trends are seen over the southern polar cap and northern Canada, with warming trends over
southern Africa, around the equator and over Europe and Russia. NH winter cooling trends



are strongest over eastern Russia and North America, but warming trends remain over
Europe.

There are a number of limitations and assumptions made for these derived trends:

i) there are only 14 years from which to extract a solar cycle component, ii) a solar cycle
component is computed for each grid box. The zonal means calculated are generally within
2 K/100 sfu of other reported solar response coefficients, but there is a strong latitudinal
and seasonal dependence (strongest solar flux response in mid-latitude winter hemisphere
– near zero response in high latitude summer), iii) we have assumed no lag between solar
flux variations and the temperature response, whereas previous work for the Davis response
for example indicates a ~160 day lag is optimal (French and Klekociuk, 2011) and iv) for
comparison with other hydroxyl temperature long-term trends we assume the global OH
layer height is well matched with the Aura/MLS 0.0046 hPa level.

To address uncertainties about the solar response coefficient we have recalculated

the global trends assuming a fixed response for each grid box (4.2 K/100 sfu derived from
the Davis observations) and also as zonal means but for a lag of 160 days (F10.7 leads T)
as previously found for Davis. These plots are available in the supplementary material and
show that, by and large, the warming and cooling patterns observed in Figure 6 do not
change significantly for the different solar cycle components.

While the WACCM-X results presented by Qian et al. (2019) are in reasonable

agreement with the OH temperature behaviour measured at Davis Station, the zonally
averaged pattern of solar cycle response and linear trend obtained from WACCM-X differs
considerably from that obtained from an analysis of the Aura/MLS data at the 0.00464 hPa
level shown in Figure 6.  In the Aura/MLS results, the solar response in both hemispheres
in winter show a great deal more variation as a function of latitude than is evident in the
WACCM-X results at 87 km (Figure 4 of Qian et al., 2019).  The zonally averaged





Aura/MLS pattern shows maxima in southern mid-latitudes in the Southern Hemisphere
(SH) winter, while the maximum is in northern mid-latitudes in the Northern Hemisphere
(NH) winter. The solar cycle response is essentially zero at 82° north and south during the
NH winter months, but it is of the order of 3 K/decade at 82° south in SH winter. The
southern hemisphere winter months have the largest variation with a pronounced maximum
in the latitude range ~10° S to 40° S. (The maximum also shows longitudinal structure
with a much broader maximum between 90° east and 90° west which is centred at higher
southern latitudes.)
The WACCM-X long term trend is predominantly negative or zero at the altitude
of the OH layer (87 km) at all latitudes and in all months apart from February, November
and December, when a positive trend of up to ~3 K/decade is present at high southern
latitudes. Aura/MLS results also show a predominantly slight negative trend ~0.5-1
K/decade, except at the equator, and at mid-latitudes in the SH winter months.
Solomon et al. (2018) simulated the anthropogenic global change through the entire
atmosphere using WACCM-X in a free-running mode (i.e., lower atmosphere below 50
km not constrained by ECMWF reanalysis data) using constant low solar activity
conditions. They find substantial cooling in the mesosphere of the order of -1 K/decade,
increasing to -2.8K/decade in the thermosphere. Temperature decreases were small near
the mesopause compared with the variation in the annual mean thus making trends there
somewhat uncertain. Solomon et al. (2018) conclude that inconsistent observational results
in the mesopause region, together with little or no global mean trends is due to the
dominance of dynamical processes in controlling mesopause temperature, which exhibits
significant interannual variability, even without variable solar forcing.
The SABER dataset (2002-2015) was used by Tang et al. (2016) to study the
response of the cold-point temperature of the mesopause (T-CPM) to solar activity. The

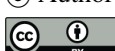



results showed that the T-CPM is significantly correlated to solar activity at all latitudes,
and the solar response becomes stronger with increasing latitude.   The solar-cycle
dependence of the mesopause cold point temperature (T-CPM) is due to the relative
importance of $CO_2$ and NO infrared cooling (Tang et al., 2016).   NO density at solar max
is about three times that at solar minimum.   Consequently, $CO_2$ cooling is relatively less
important at solar maximum, but is the dominant cooling mechanism during solar
minimum.

Values of the solar response of T-CPM reported by Tang et al. (2016) increased

from 2.82 ± 0.73 K/100 sfu at 0-10° S to 6.35 ± 1.16 K/100 sfu at 60-70° S.   Correlation
coefficients of mesopause temperature with F10.7 cm solar irradiance data were higher for
mid-latitudes (> 0.9) than at the equator (~0.7) and at higher latitude.   The value found for
70° S (~0.8) is consistent with the correlation coefficient obtained for the OH*
temperatures (Figure 1(a) $(0.584)^{1/2}$ ~ 0.76) obtained in this work.   At low latitudes, one
would expect the QBO and ENSO to be significant factors there (see e.g., Nath and
Sridharan, 2014), but at high latitudes, gravity wave activity is a candidate for the missing
variance.   Inter-annual variations of GWs at high latitudes are correlated with the strength
of the polar vortex.   A stronger polar vortex filters out more eastward propagating GWs,
thus leading to more westward GW drag, which drives stronger meridional circulation
(Karlsson and Shepherd, 2018).

Although the altitude of the cold point changes with season (e.g., Yuan et al., 2019)

it tends to be higher than the centroid of the OH* layer, the global solar response value
obtained for T-CPM (4.89 ± 0.67 K/100 SFU) is in good agreement with the solar response
coefficient derived from ground-based OH* observations.

The solar response of the T-CPM in Tang et al. (2016) shows some significant

differences from the results in Figure 6 (zonal mean cycle from Aura/MLS) of this work.



The solar response of the T-CPM increases more or less monotonically with latitude,
whereas the solar response registered by Aura/MLS maximises at higher mid-latitudes. Of
course the height of the T-CPM is some 7 km higher on average as indicated in Figure 9
(b) of Tang et al. (2016).

Several authors (Perminov et al., 2014; Pertsev and Perminov, 2008) have reported

that winter OH* temperatures are more sensitive to the solar flux variation than summer
temperatures.  This agrees with the Aura/MLS variation shown in Figure 6.

As a final comment on the global trends, it is noted that the largest errors in the

linear trend fit for the SH winter occur coincident with the regions positive or negatively
correlated with the QQO (cf. figure 3. i.e., eastern Antarctic polar cap, southern Pacific and
southern Indian oceans). This is understandable if there is a significant QQO signal
superposed on the underlying long-term linear trend.

## 700    6. Summary and Conclusions

We provide updates for the long-term trend and solar cycle response derived from

24 years of spectrometer observations of hydroxyl airglow at Davis Research Station,
Antarctica (68° S, 78° E). A cooling trend in the mean winter temperatures [D106-259] of
-1.20 ± 0.51 K/decade (95% confidence limits -0.14 K/decade < L < -2.26 K/decade) is
obtained coupled with a solar cycle response coefficient of 4.30 ± 1.02 K/100sfu (95%
confidence limits 2.2 K/100sfu < S < 6.4 K/100sfu). The observed cooling is consistent
with radiative cooling due to increasing $CO_2$ concentrations and a rate of -0.06 K/ppm$CO_2$
or -0.22 K/%$CO_2$ is implied (ignoring possible contributions of stratospheric ozone change
to the trend). A significant note is that a new record low winter-mean temperature was set
for the Davis measurements in 2018, with a value of 198.3 K, which is 1.7 K below the



previous minimum recorded in 2009 (200.0 K). An examination of the seasonal variation
in the trend fit parameters reveals very little (no significant) long-term trend occurs over
the 2 midwinter months of June and July, but 95% significant trends of -1.5 to -2.6
K/decade during the April-May and August-October intervals. From examination of
TIMED/SABER VER profiles we see no evidence that the trend results obtained can be
significantly attributed to a change in the height of the OH layer.

We do not see evidence of a trend break or a change in the nature of the underlying

trend after accounting for the solar cycle response in the Davis OH temperatures, however,
this simple solar-cycle and linear trend model fit accounts for only 58% of the temperature
variability. The remaining variability reveals evidence of a temperature oscillation on a
quasi-quadrennial (~4 year period) timescale.

We compare our observations with Aura/MLS version v4.2 level 2 data over the

last 14 years when these satellite data are available and find close agreement (a best fit)
with the 0.00464 hPa (native Aura/MLS retrieval) pressure level values. The solar cycle
response, long-term trend and underlying QQO residuals are consistent with the Davis
observations.  Consequently, we derive global maps of Aura/MLS trend and solar response
coefficients for the SH and NH winter periods to compare with other observers and models.
Significant patterns for the zonally averaged solar cycle response are maxima in southern
mid-latitudes in the Southern Hemisphere (SH) winter and in northern mid-latitudes in the
Northern Hemisphere (NH) winter. Long term trends are a predominantly slight negative
(~0.5-1 K/decade), except at the equator, and at mid-latitudes in the SH winter months.
Comparisons are also made with the WACCM-X model and mesopause cold point
temperature versus solar activity study using TIMED/SABER data of Tang et al. (2016),
both of which reveal significant differences in the zonally averaged patterns of solar cycle
response and linear trend compared to the Aura/MLS data at 0.00464 hPa.

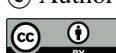



Further analysis using the datasets described here are undertaken to examine the
residual QQO signal that this analysis has revealed. A second part of this paper "Analysis
of 24 years of mesopause region OH rotational temperature observations at Davis,
Antarctica. Part 2: Evidence of a quasi-quadrennial oscillation (QQO) in the polar
mesosphere." concerns this observation.

## 742 Data Availability

All Davis hydroxyl rotational data described in this manuscript are available through the
Australian Antarctic Data Centre website (ref project AAS4157) via the following link
https://data.aad.gov.au/metadata/records/Davis_OH_airglow . The satellite data used in
this paper were obtained from the Aura/MLS data centre (see https://mls.jpl.nasa.gov), the
SABER data centre (see http://saber.gats-inc.com/data.php) and are publicly available.

## 749 Author Contribution

WJRF managed data collection, performed data analysis, prepared manuscript with
contributions from all co-authors
FJM analysis of SABER data, manuscript editing, figures, references
ARK analysis of Aura/MLS satellite data, manuscript editing.

## 755 Competing Interests

The authors declare that they have no conflict of interest.



## Acknowledgements

The authors thank the dedicated work of the Davis optical physicists and
engineers over many years in the collection of airglow data and calibration of
instruments. This work is supported by the Australian Antarctic Science Advisory
Council (project AAS 4157).
The satellite data used in this paper were obtained from the Aura/MLS data centre
(see https://mls.jpl.nasa.gov), the SABER data centre (see http://saber.gats-
inc.com/data.php) and are publicly available. We thank those teams and acknowledge the
use of these data sets.
This work contributes to the understanding of mesospheric change processes
coordinated through the Network for Detection of Mesospheric Change (see
https://ndmc.dlr.de/)

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
