# Peer review of "Analysis of 24 years of mesopause region OH rotational temperature"

_Atmospheric Chemistry and Physics, 2019_

## Referee Comment (RC1) · Anonymous Referee #2 · 4 Feb 2020

Review of "Analysis of 24 years of mesopause region OH rotational temperature observations at Davis, Antarctica. Part 1: Long-term trends" by French et al.

This manuscript presents the analysis of a very long dataset of OH temperatures over Antarctica. This is an extension of 8 years of the dataset presented by French and Klekociuk (2011). Indeed, results for the trends derived here coincide with their previous results and those for the solar response are only slightly different. In this work, the authors further identify and isolate a close-to-4-year period signal, which is to be studied in the second part of this work. Even if the results in this paper may initially look as a mere update of previous results using an extended database, they are interesting and certainly worth publishing because they show the persistence of the trend and the consistency of the solar signal. Therefore, I suggest the publication of this paper in ACP, once the following concerns and comments are taken into account.

Line numbering corresponds to the latest version uploaded by the authors (acp-2019-1001-manuscript-version3.pdf).

Main comments

There is no discussion on the effect of MLS broad vertical resolution in the mesosphere and the potential impact on the comparisons shown in the manuscript. Indeed, it would be interesting to see comparisons with SABER, even with a smaller winter temporal-coverage. Additionally, SABER provides information on the altitude of the OH layer, potentially providing a more accurate approach. In the same context, the choice of a fixed pressure level in MLS data (seemingly done based just on a better agreement) is not very well justified, as the layer altitude varies.

There is also a lack of discussion on previously reported seasonal or latitudinal effects on trends that the authors mention but do not connect with their results. It would be useful to overplot these results from other authors on the corresponding figures in the manuscript.

The discussion section is too long. It is a good review but it is not easy to follow and, more importantly, to see how the results presented here fit on the discussion. I suggest revising the section, shortening it and putting the results of this paper into the context.

Other comments and suggestions are:

L42-45 Include in the abstract the result of the global MLS trend analysis.
L148-L151. Is there any error associated to that interpolation?
L151. Is that 2% contribution independent of temperature? Or in other words, do potential uncertainties in the Q-line contribution incur into significant errors in the derived temperatures?
L157. Please, write a short sentence explaining why your choice is Langhoff et al. (1986). Do you reach better agreement with satellites when using specific values?
L160. Did you explicitly test the effect on trends of the probabilities used? How much is "not significantly"? This is important in order to understand differences in trends between the different datasets.
L161-163. This sentence is somehow redundant. If biases due to the choice of probabilities can reach 12K, it is obvious that comparisons between different instruments depend on the choice. Please, delete.
L169. Specify selection criteria quantitatively.
L183 and Fig. 1. There are more recent versions for MSISE. Use latest version or at least show that it makes no difference.
L190. Introduce Fig. 2 at the beginning of this paragraph.

L207-210. Remove this text from the caption. It is already in the text

L228. Are these MLS nightly means? Is there any time difference criterion used? If not, consider discussing possible sampling effects.

L246-252 and Fig. 4. This is a very nice and colorful figure but is it really essential or a short sentence would be enough?

L284. The choice of this altitude range sounds somehow arbitrary. The OH layer is not centered at 85km. Please, justify. What happens if you use 85-90km?

L290. This is the main problem I see using MLS data for this study. Why don't you use SABER data, even with a smaller time-coverage during the winter months? It could do a good job in your 60-day running means solar and linear trends. Garcia-Comas et al. (2014) showed that MLS biases at these altitudes could be large on the South Pole.

L293. The choice of this pressure level is not justified. You could be getting a good agreement due to a bias that could be masking a wrong selection of altitudes. Indeed, it is well known that the altitude of the OH-layer is variable (as you even mention in L582-584). Please, discuss this point.

L299. Yes, you show this very nicely in Fig. 5, my favorite figure of the paper.

L302. Indicate the figure where this is shown.

L304. Aura/MLS trends "at 0.00464 hPa"

L307. How did you derive that this is the OH equivalent pressure level and that it does not change with latitude? Does SABER, measuring OH emission and temperature vs z and pressure, show a significant change of equivalent pressure with latitude?

L308. Monthly anomalies? 60-night running means? What are these?

L316. What is the origin of these enhance bands?

Section 5.1. I enjoyed reading the review but it could probably be shortened and better organized and your results should be put into the context you describe. Are they reasonable? Do they agree? Does the seasonality of your data agree with other results? Does it agree with the expected variations? I suggest extending the title of this section in order to include ozone.

L440. Include reference.

L449. Include reference.

L465. This is already said, also mentioning the same reference.

L478-486. It is not clear to me what this has to do with this work. Hervig et al. (2019) paper mainly deals with the paradox on the solar response of H2O. Perhaps mentioning only their result related to temperature makes more sense. By the way, do you actually see a change of solar response of temperature from 1995 to 2018?

L488. Also Solomon et al. 2018

L501-502 I do not understand "global averaged temperature (..) as a function of latitude". How does their month-to-month variability compare with the seasonal variability you derive? Please, overplot on Fig. 5 and discuss.

L507. Please, write altitude

L559. Write "from 1995 to 2018". If you remove your QQO, don't you see such breaks? From your measurements, it seems that your trend is not monotonic. Quantify "no obvious sign".

L584. Please, also mention Liu et al. (2006).

L592. Garcia-Comas et al. (2017) also estimated the trend and solar response of OH* altitude and temperature from SABER.

P610. What is the expected change in MLS temperatures due to a change of 0.02-0.04 km? This may lead to a bias in the comparison between DAVIS and MLS. A way to test this could be done with SABER data by comparing temperature trends at a fixed altitude to temperature trends at a OH VER weighted temperature.

L657-661 Why are these results so different? Could the difference be due to sampling?

L720. According to a possible contribution of 30% by ozone, these values have (at least) a 30% error.

L723-726. Could you provide the reason for this minimum in 2009?

L747. Mention here also the results from other OH observations (those listed in Table 1).

Fig1b. Do all years contain the same days of measurements from doy 106 to 259? If not, there could be a sample bias.

Figure 2. Time coverage changes with doy. What is the effect of DW1?

Fig 5.a. the minimum in solar trend is during the month when downwelling is maximum. This migh be an indirect compensation of the cooling due to the direct dependence of downwelling (warming) and solar flux (COMPROBAR!!!)

Fig 5b. Perhaps you might be sounding different altitudes? What is the seasonal change of the altitude of the OH layer? Did you look at SABER data? Also, this might be connected to O3 trend seasonality or CO2 trend seasonality.

Figure 5. Define grey boxes and blue dots.

Figure 6. What is time sampling for MLS? Are you removing tides? Trends strongly depend on sampling (Rezac et al. 2018)

Fig 6. Please, overplot trend at Davis on the 1d plot. Also indicate CAP and LEO position on the maps.

Fig 6. b. The blue/red bands at 70N in the NH winter months look like the trend and solar response related to stationary PWs.

Fig. 7. What is the realtionship between this plot and the temperature anomaly? Can it help to explain differences between DAVIS and MLS?

Table 1. Discuss these results in the text, particularly mention them in section 5.

Table 1. Include MLS results in this list

---

## Referee Comment (RC2) · Anonymous Referee #1 · 7 Feb 2020

Reviewer Report on the manuscript acp-2019-1001

Analysis of 24 years of mesopause region OH rotational temperature observations at Davis, Antarctica – Part 1: Long-term trends

by W.John R. French et al.

General Remarks

1...The paper presents 24 years of observations of OH temperatures, which is an intersesting extension of an earlier data set worth publishing.

2 The data were taken in Antarctica where such measurements cannot be performed

in summer. This is a drawback for several interpretation aspects and must be carefully considered.

3 The data are discussed in the context of increasing CO2 mixing ratios. They are extensively compared to MLS and SABER satellite results, and to computer models (WACCM-X).

4 The paper gives a long term analysis and discusses possible trend breaks. These results are questionable because of the lack of winter data.

5 The authors see a quasi-quadrennial oscillation (QQO) in their data. They announce a detailed discussion in a second part of the paper. This should take into account recent work in the literature on 3 – 5 year oscillations.

6 The paper is well written, and is recommende for publication after some modifications.

Major Comments

Line 221 pp: Figure 3 indicates five oscillation periods. The approximate period lengths are 2x 3 yr, 1x 4 yr, and 2x 5 yr. It is not obvious that a mean can be taken. Superposition of a 3 yr and a 5 yr oscillation should be checked (see for instance Offermann et al., JASTP 135,1, 2015).

Line 383 pp, 540: The paper Offermann et al., 2006, should not be used to demonstrate a trend break. It was outdated by Offermann et al., JGR 115, D18127, 2010, who show a longer data series.

Line 405 pp, Section 5.1, 5.4: In the discussion of the trend data it should be elaborated that the summer data at Davis are missing. Trend data are different in summer and winter as shown by the MLS data in your Fig. 6. They can also vary from month to month as shown in your Fig. 5, but variations could be much larger (see for instance Offermann et al., 2010, their Fig.9). Possibly the summer trends are larger than your 1.2 K/decade (by number), and so might be the trend of annual data. Hence, if you

want to include Davis data to Tab.1 please use annual MLS data!

Line 558/559: ..."no ...sign of a discontinuity in the trend..." Kalicinsky et al., 2018, in their long-term analysis find that the summer data may be much more important than the winter data. Therefore please check your above statement by means of annual MLS data.

Minor Comments

Line 219: The error of your solar cycle response (1.02 K/100sfu) appears relatively large (see Tab.1). Do you know a reason?

Line 295: Please give the error of the MLS trend.

Line 325 pp, Fig.6: a) Please show Panel numbers. b) Please show latitude scales. c) Part of the captions are difficult to read. d) Line 334: There is no Fig.1B. Do you mean Fig.3b?

Line 353 pp, Table 1: Plese give the selection criteria for the sires shown.

Line 456 pp: "...peak altitudes.." Here and in the following it is sometimes unclear whether you mean the maximum of the peak or the geometric altitude. Please clarify.

Line 505 pp, 508: It is unclear whether you mean your Fig.5 or Qqian et al.. Please clarify.

Line 686: Fig.1a does not show this! Do you mean that you derived it from this Figure?

Line 693: Sentence difficult to understand.

Line 706 pp: Where can this be seen?

The Supplementary Material was not available to me.

---

## Author Comment (AC1) · 6 Apr 2020

Reviewer Report on the manuscript acp-2019-1001 Analysis of 24 years of mesopause region OH rotational temperature observations at Davis, Antarctica – Part 1: Long-term trends by W.John R. French et al.

General Remarks

1. . .The paper presents 24 years of observations of OH temperatures, which is an intersesting extension of an earlier data set worth publishing.

Thank you.

2 The data were taken in Antarctica where such measurements cannot be performed in summer. This is a drawback for several interpretation aspects and must be carefully considered.

Observations of the hydroxyl nightglow cannot be made over the summer at this latitude and we understand this is a limitation of the observational program for long-term trends using this technique. We contribute a solar and long-term trend assessment of the mean winter temperatures at this high southern latitude site and make comparisons with satellite observations to place these observations into global context.

3 The data are discussed in the context of increasing CO2 mixing ratios. They are extensively compared to MLS and SABER satellite results, and to computer models (WACCM-X).

Yes, comparison with other observations and models place these measurements into context.

4 The paper gives a long term analysis and discusses possible trend breaks. These results are questionable because of the lack of winter data.

Winter data is provided. It is the southern hemisphere *summer* data that is lacking. We make this clear from the outset of the manuscript.

5 The authors see a quasi-quadrennial oscillation (QQO) in their data. They announce a detailed discussion in a second part of the paper. This should take into account recent work in the literature on 3 – 5 year oscillations.

The second part of this work is available in discussions as acp-2019-1097

6 The paper is well written, and is recommende for publication after some modifications.

Thank you.

Major Comments

Line 221 pp: Figure 3 indicates five oscillation periods. The approximate period lengths are 2x 3 yr, 1x 4 yr, and 2x 5 yr. It is not obvious that a mean can be taken. Superposition of a 3 yr and a 5 yr oscillation should be checked (see for instance Offermann et al., JASTP 135,1, 2015).

The quasi-quadrennial periodicity revealed in the residual temperatures (seasonal, solar cycle and long-term trend fits removed) is an interesting feature and forms the basis of part 2 of this work; available in discussions as **acp-2019-1097**. We examine many possible sources for this feature using

correlation and composite analyses with other data sets. We cannot ascertain whether the oscillation is a superposition of 3 and 5 year periodicities.

Line 383 pp, 540: The paper Offermann et al., 2006, should not be used to demonstrate a trend break. It was outdated by Offermann et al., JGR 115, D18127, 2010, who show a longer data series.

The reference to Offermann et al., 2006 applied to a time period when trend breaks first began to appear in the literature in relation to mesopause region temperature trends. In that context the reference is still valid despite it being later updated. We have added the updated reference (Offermann et al., 2010) to the paragraph noting the continued occurrence of trend breaks.

Offermann, D., P. Hoffmann, P. Knieling, R. Koppmann, J. Oberheide, and W. Steinbrecht (2010), Long-term trends and solar cycle variations of mesospheric temperature and dynamics, J. Geophys. Res., 115, D18127, doi:10.1029/2009JD013363

Line 405 pp, Section 5.1, 5.4: In the discussion of the trend data it should be elaborated that the summer data at Davis are missing. Trend data are different in summer and winter as shown by the MLS data in your Fig. 6. They can also vary from month to month as shown in your Fig. 5, but variations could be much larger (see for instance Offermann et al., 2010, their Fig.9). Possibly the summer trends are larger than your 1.2 K/decade (by number), and so might be the trend of annual data. Hence, if you want to include Davis data to Tab.1 please use annual MLS data!

The fact that we derive a trend in the mean *winter* temperatures is explicitly stated in the first sentence of the discussion in section 5.1 (lines 407-408) and in the first sentence of section 5.4 (line 619) "southern hemisphere (SH) *winter* months (AMJJAS)". We think it should be clear to the reader that hydroxyl temperatures at Davis cannot be obtained over the summer months as the sun does not go down.

We do explore the seasonal variation in trends where possible over the observing season at Davis (fig 5) and understand that the trends are variable.

We also compare with MLS over winter and summer for each hemisphere (Figure 6) to show the seasonal difference in trends. MLS does not indicate that the summer trends [ONDJFM average trend plots in Fig 6] are larger than -1.2 K/decade for the grid box over Davis. The grid box over Davis yields a long-term trend of +0.02±0.08 K/decade for the summer months [ONDJFM] and an annual average value [JFMAMJJASOND] of -0.37±0.06 K/decade.

Table 1 is a list of ground based observations. It does not contain the MLS trends as they are globally and seasonally variable. Figure 6 shows the map of these coefficients for comparison with the ground based observations in Table 1.

Line 558/559: . . ."no . . .sign of a discontinuity in the trend. . ." Kalicinsky et al., 2018, in their long-term analysis find that the summer data may be much more important than the winter data. Therefore please check your above statement by means of annual MLS data.

MLS values for the summer and winter seasons for each hemisphere are shown in Figure 6. It does not appear that the summer values are greater than winter values for the MLS trend coefficients globally (the two linear trend plots in figure 6 use the same colorbar scale). Coefficients maps (linear trend and solar cycle) for the whole year (all month MLS averages) are provided below for comparison with those in Fig 6. It merges the winter and summer features (as one would expect) and particularly highlights the mid-latitude maxima in solar cycle response.

MLS Linear trend coefficient [JFMAMJJASOND]

[Figure]

MLS Solar cycle coefficient [JFMAMJJASOND]

[Figure]

Minor Comments

Line 219: The error of your solar cycle response (1.02 K/100sfu) appears relatively large (see Tab.1). Do you know a reason?

The solar cycle response error is not unreasonably large compared to others in Table 1. (less than, or of the same order as 7 out of 11). We would suggest that the main reason for the error is the goodness-of-fit of the F10.7 and linear trend model. In particular that the model does not contain a quasi-quadrennial oscillation term that is evident in the residuals.

Line 295: Please give the error of the MLS trend.

MLS trend is 1.4±1.1 K/decade . added to text

Line 325 pp, Fig.6: a) Please show Panel numbers. b) Please show latitude scales. c) Part of the captions are difficult to read. d) Line 334: There is no Fig.1B. Do you mean Fig.3b?

Added panel numbers and latitude scales and modified the figure to improve readability. Yes, thank you we mean figure 3b for the comparison, but the caption has changed with the addition of all sites to the map for comparison as requested by another reviewer.

Line 353 pp, Table 1: Plese give the selection criteria for the sires shown.

We have updated the table from Table 2 in French and Klekociuk (2011) where updates were available since 2011.

Line 456 pp: ". . .peak altitudes.." Here and in the following it is sometimes unclear whether you mean the maximum of the peak or the geometric altitude. Please clarify.

We mean the altitude of the peak of the VER profile. This section has been extensively modified in response to another reviewer and this paragraph is no longer included.

Line 505 pp, 508: It is unclear whether you mean your Fig.5 or Qqian et al.. Please clarify.

This refers to the seasonal variation in the solar cycle coefficient in our Fig 5(a) and can be compared with Fig 4 of Qian et al (2019); this is clarified in the text.

Line 686: Fig.1a does not show this! Do you mean that you derived it from this Figure?

The $R^2$ value is shown in Fig 3(a) (Fit of solar cycle and long-term trend model to OH temperatures). This is corrected in the text.

Line 693: Sentence difficult to understand.

Modified sentence to read "Although the altitude of the mesospheric cold point changes with season (e.g., Yuan et al., 2019) and tends to be higher than the centroid height of the OH* layer, the global solar response value obtained for T-CPM (4.89 ± 0.67 K/100 SFU) is in good agreement with the solar response coefficient derived from ground-based OH* observations."

Line 706 pp: Where can this be seen? The Supplementary Material was not available to me.

The original Fig 6. also contained a plot of the error in fitting the solar cycle and long term linear trend model (see below) and the point was made here that where the error was largest coincides with a strong QQO signal. However the QQO investigation is now discussed entirely in Part 2 of this work (acp-2019-1097).

The paragraph was modified to read "As a final comment on the global trends, it is noted that the largest errors in the linear trend fit for the SH winter understandably occur coincident with the regions positively or negatively correlated with the QQO (not shown here). The fit can be significantly improved if the QQO component can be understood and modelled. We investigate the QQO in detail in part 2 of this work. "

**Error in model fit to MLS [AMJJAS]**

[Figure]

Error in Residual Fit Slope (K/decade)

---

## Author Comment (AC2) · 9 Apr 2020

**Response to RC1**

Review of "Analysis of 24 years of mesopause region OH rotational temperature observations at Davis, Antarctica. Part 1: Long---term trends" by French et al.

This manuscript presents the analysis of a very long dataset of OH temperatures over Antarctica. This is an extension of 8 years of the dataset presented by French and Klekociuk (2011). Indeed, results for the trends derived here coincide with their previous results and those for the solar response are only slightly different. In this work, the authors further identify and isolate a close---to---4---year period signal, which is to be studied in the second part of this work. Even if the results in this paper may initially look as a mere update of previous results using an extended database, they are interesting and certainly worth publishing because they show the persistence of the trend and the consistency of the solar signal. Therefore, I suggest the publication of this paper in ACP, once the following concerns and comments are taken into account.

Line numbering corresponds to the latest version uploaded by the authors (acp---2019--1001---manuscript---version3.pdf).

Thank you for your considered and detailed review of acp-2019-1001. We address your comments and suggestions below.

Main comments

There is no discussion on the effect of MLS broad vertical resolution in the mesosphere and the potential impact on the comparisons shown in the manuscript. Indeed, it would be interesting to see comparisons with SABER, even with a smaller winter temporal--coverage. Additionally, SABER provides information on the altitude of the OH layer, potentially providing a more accurate approach. In the same context, the choice of a fixed pressure level in MLS data (seemingly done based just on a better agreement) is not very well justified, as the layer altitude varies.

The broad vertical resolution (15km averaging kernel) of MLS profiles in the mesopause region is referenced on line 290. Its effect is to integrate temperatures from above and below the OH layer into the computed temperature for that pressure level. Bearing in mind that the OH profile is itself a broad layer (FWHM ~8km) and the rotational temperatures computed correspond to a similar integration over the width of the layer it is not unreasonable to compare OH with MLS temperatures. Indeed we find good agreement.

We have previously reported, and routinely compare our measured OH temperatures with both Aura/MLS and SABER profiles. In particular, French and Mulligan, 2010 examined biases between Davis OH temperatures and both Aura/MLS and SABER. A significant limitation of SABER for comparisons with Davis observations is the yaw cycle sampling of the satellite. Comparable observations over Davis are confined to two intervals (day-of-year 75–140 and 196–262) and days prior to 106 and after 259 are outside the OH winter averaging interval. Therefore only days 106-140 and 195-259 are comparable and a large part of the winter months is not sampled. As a consequence SABER winter averages do not fit the OH observations at Davis as well as MLS.

We agree that the geometrical altitude of the layer varies. Since the hydroxyl layer position is primarily controlled by collisional quenching with $O_2$ and $N_2$ on the bottom-side of the

layer, and reaction with atomic oxygen on the top-side of the layer it is the concentration (density) of the reacting species that governs the layer position. Therefore it is reasonable to compare with MLS pressure (proportional to density) levels than on geometrical altitude levels. See also details of SABER altitude and pressure plots addresses in item 20 below.

This work is primarily concerned with mesopause region trends and absolute temperature biases are removed by subtraction of the climatological mean. We work here with anomalies (difference from the mean of all years) and residuals (solar cycle component removed). In selecting the 0.0046hPa level we compared the Davis OH winter average anomaly with MLS [AMJJAS average] anomaly over a range of altitude and pressure levels. (see plots below, the first are altitude ranges, the second pressure levels, both temperatures and the anomaly are shown).

[Figure]

[Figure]

Altitude ranges 78-83km, 83-88km and 85-90km and pressure ranges 0.00215hPa and 0.00464hPa are all in reasonable agreement (<5K in absolute terms) with the OH temperatures, but we know there are biases with MLS (see French and Mulligan, 2010), and we know that the Davis OH temperatures are ~2 K high using LWR transition probabilities compared to those computed with the experimentally measured transition probability ratios determined in French et al 2000. These biases are removed by comparing anomalies. We calculate the Chi-Square goodness of fit parameter between the OH winter average *anomaly* with the Aura/MLS anomalies. The 0.0046hPa pressure level yields the smallest chi-sqr (14.8) compared to a layer centred on the traditional 87km altitude level (85-90km chisqr=18.8). This difference is small, but we prefer the pressure level comparison for the reason given above.

The relationship between pressure and geopotential height (GPH) is examined below using the MLS data set. Global decreases in GPH anomaly (between ~110 to 240 metres/decade) at the 0.0046hPa pressure level are consistent with a contraction of the underlying

atmosphere and also consistent with the SABER trend in OH mean winter layer altitude for Davis shown in Fig 7. (200metres/decade) and discussed in the text.

MLS slope of fit to residual geopotential height at 0.0046hPa

[Figure]

There is also a lack of discussion on previously reported seasonal or latitudinal effects on trends that the authors mention but do not connect with their results. It would be useful to overplot these results from other authors on the corresponding figures in the manuscript.

We have modified figure 6 to indicate the solar cycle response and long-term trend coefficients available from other authors listed in Table 1.

A comparison of the seasonal variations in long-term trends has been previously published in a similar figure to Fig 5. in French and Klekociuk, 2011 (their figure 7; reproduced below). This included the results of Offermann et al 2010 at Wuppertal and Espy and Stegman (2002) from Stockholm. Since there has not been further updates to the seasonal variation in trend coefficients at either site we have not replicated this comparison. Perminov et al., 2018 offer seasonal variances in OH temperatures for the Russian sites of Zvenigorod and Tory but do not compute seasonal trend components.

[Figure]

**Figure 7.** The 30 day sliding window (5 day step) evaluations of (a) solar cycle and (b) long-term trend coefficients at zero F10.7 lag (i.e., the vertical transect through Figure 6 (right-hand panels) at zero lag). One-sigma error bars and the 95% confidence limits (upper and lower traces) are as marked. The (true calendar) monthly evaluations are also plotted and labeled. Overlaid on Figure 7b are the equivalent monthly trend results by *Offermann et al.* [2010] from Wuppertal (labeled WUP) and *Espy and Stegman* [2002] from Stockholm (labeled STO). These are approximate values scaled off their respective seasonal trend plots and are offset by 6 months to match the Southern Hemisphere season.

The discussion section is too long. It is a good review but it is not easy to follow and, more importantly, to see how the results presented here fit on the discussion. I suggest revising the section, shortening it and putting the results of this paper into the context.

The discussion section has been substantially revised and some sections removed.

Other comments and suggestions are:

1. L42‑‑‑45 Include in the abstract the result of the global MLS trend analysis.

We have included in the abstract the Aura/MLS solar cycle (3.39 ± 2.3 K/100 sfu) and long term trend (-1.3 ± 1.2 K/decade) coefficients at Davis for comparison. These are computed from anomalies derived from the AMJJAS means of all satellite observations within 500 km of Davis station over the 14 years of MLS observations (compared to 24 years of OH observations at Davis).

We note from Fig 6 that significant variability appears in both long term trend and solar cycle coefficients computed from MLS on a global scale. The global coefficients from MLS computed in the 5° latitude x 10° longitude grid boxes for AMJJAS averages at 0.0046hPa range from -2.3 to +2.3 K/decade (mean -0.01 K/dacade) for the long term trend and -0.2

to 8.8 K/100 sfu (mean 3.3 K/100sfu) for the solar cycle. It is not practical to include all the global MLS trend results in the abstract.

2.  L148---L151. Is there any error associated to that interpolation?

Yes. The interpolation attempts to account for changes in the overall intensity of the OH emission during the course of the 7 minute scan. In some cases the intensity may not vary in a linear fashion, but in general interpolating intensities to a common time between consecutive scans provides a better estimate for varying intensity of the OH emission. Selection criteria limit extreme rates of change of intensities (<6%) between consecutive spectra.

The error assigned to each line intensity is the square root of the total number of counts, together with an error in estimating the background under each line. A standard deviation error is also derived from the 3 different ratios contributing to the weighted mean temperature for each pair of consecutive spectra. The process is described in detail in French and Burns (2004) in sections 2. Measurements and 3. Rotational temperature analysis.

French, W. J. R. and Burns, G. B.: (2004) "The influence of large-scale oscillations on long-term trend assessment in hydroxyl temperatures over Davis, Antarctica", J. Atmos. Sol. Terr. Phys., 66, 493–506. 634

3.  L151. Is that 2% contribution independent of temperature? Or in other words, do potential uncertainties in the Q---line contribution incur into significant errors in the derived temperatures?

No, the $Q_1(5)$ is not temperature independent, but its contribution to $P_1(2)$ is computed in an iterative process using the final weighted mean temperature from three possible ratios from the $P_1(2)$, $P_1(4)$ and $P_1(5)$ lines. The contribution of the two lambda doubled components of $Q_1(5)$ and computed separately from the final weighted mean temperature. Approximately 98.0% of $Q_1(5)$e and 47.5% of $Q_1(5)$f contribute to the $P_1(2)$ emission intensity we measure, depending on the instrument line shape measured via frequency stabilised laser.

4.  L157. Please, write a short sentence explaining why your choice is Langhoff et al. (1986).

We use Langhoff et al. (1986) transition probabilities because they are closest to the experimentally measured, temperature independent line ratios determined for the OH(6-2) band using the same instrument in French et al 2000.
Recent work by Noll et al ([https://www.atmos-chem-phys-discuss.net/acp-2019-1102/acp-2019-1102.pdf](https://www.atmos-chem-phys-discuss.net/acp-2019-1102/acp-2019-1102.pdf)) also show relatively small errors in the comparison of populations from P- and R- branch lines for the Langhoff et al (1986) coefficients, as well as van der Loo and Groenenboom (2008) and Brooke et al. (2016) coefficients. The latter two sets were not available at the time of that study.
The paragraph was modified to encompass this explanation.

5.  Do you reach better agreement with satellites when using specific values?

This study is not an assessment of the bias between Davis OH temperatures derived with different transition probabilities and satellite measurements. We have previously examined this in French and Mulligan, 2010.

This work concentrates on the trends and variability inherent in the annual anomalies (any bias is removed by subtraction of the climatological mean) and thus is independent of the choice of transition probabilities (see further comments below). However, we do obtain good agreement of absolute temperatures using Langhoff et al (1986) probabilities (justified above) with Aura/MLS at the 0.0064hPa level and with SABER using a Gaussian fit to the OH-B channel VER (French and Mulligan, 2010), given the many assumptions made with regard to the layer height and shape, the width of the averaging kernel used for satellite retrievals.

6. L160. Did you explicitly test the effect on trends of the probabilities used? How much is "not significantly"? This is important in order to understand differences in trends between the different datasets.

From the rotational temperature equation for the temperature derived from the ratio of emission lines $m$ and $n$

$$T_{rot} = \frac{E_m - E_n}{k ln\left(\frac{I_n.A_m.(2J'_m + 1)}{I_m.A_n.(2J'_n + 1)}\right)}$$

Where $E$ are the upper state energies, $I$ are the measured intensities, $A$ are the transition probabilities, $J'$ are the upper state rotational quantum numbers and $k$ is Boltzmann's constant.

Choice of a particular transition probability set only affects the ratio $A_m/A_n$ and corresponds to an offset in $T_{rot}$. While this choice is important for comparisons of absolute temperature observations between sites, it is not important for studies of trends and variability so long as the same transition probability set has been used consistently for all years.

In this study, removal of the climatological mean, subtracts any offset due to differences in the transition probability ratio. The only conceivable differences between the temperatures derived using different sets is selection criteria boundary effects (whether individual measurements pass selection criteria on the extrema of the selection criteria limits). We believe (as for item 4 above) that Langhoff et al (1986) TP's are consistent with the experimentally measured, temperature independent ratio's examined in French et al 2000 and thus provide a reasonable estimate of the absolute temperature for comparison with SABER, Aura MLS, and other observations, however we make no assessment of bias against other observations here. Rather we assess trends and variability in the anomalies. The paragraph has been modified to express these points.

7. L161---163. This sentence is somehow redundant. If biases due to the choice of probabilities can reach 12K, it is obvious that comparisons between different instruments depend on the choice. Please, delete.

Deleted.

8. L169. Specify selection criteria quantitatively.

Paragraph modified to read "Selection criteria limit extreme values of weighted standard deviation (< 20 K) and counting error (< 15 K), slope (< 0.06 counts/Å), magnitude (< 250 counts per second) and rate of change (< 3 counts per minute) of the backgrounds and the rate of change of branch line intensities (< 6%) between consecutive scans. Further details of the rotational temperature analysis procedure are available in Burns et al. (2003) and French and Burns (2004)"

9. L183 and Fig. 1. There are more recent versions for MSISE. Use latest version or at least show that it makes no difference.

Reference model updated to NRLMSISE-00. It makes little difference on the scale of the plot in fig 1.

[Figure]

Fig1 caption modified accordingly and the reference updated to Picone et al 2002 [Picone, J. M., A. E. Hedin, D. P. Drob, and A. C. Aikin, NRLMSISE-00 empirical model of the atmosphere: Statistical comparisons and scientific issues, J. Geophys. Res., 107(A12), 1468, doi:10.1029/2002JA009430, 2002.]

The reference atmosphere (values for local midnight at Davis) has also been added to Figure 2 for comparison and text added to discuss the differences.

[Figure]

Figure 2. Superposed nightly mean temperatures from 1995 to 2018 [gray points] and a 5-day running mean which represents the climatological mean [orange line] with 1σ intervals [black lines]. The seasonal variation [green annual, semi-annual, ter-annual fit] and mid-April and mid-August dips [red arrows] are also indicated. Green vertical lines mark the calculation region for winter mean temperatures (inside the winter to summer transition intervals). The NRL-MSISE00 reference atmosphere (local midnight values for Davis) is also added for comparison [gold points]

10. L190. Introduce Fig. 2 at the beginning of this paragraph.

Inserted "(Fig. 2)" at the end of the first sentence and removed the sentence introducing Figure 2 midway through this paragraph.

11. L207---210. Remove this text from the caption. It is already in the text

Removed text and modified caption as for item 9 above.

12. L228. Are these MLS nightly means? Is there any time difference criterion used? If not, consider discussing possible sampling effects.

These are AMJJAS (6-month) means derived from all MLS observations within 500 km of Davis station as described in that line and in the paragraph from L282. There are only about 60 coincident samples per month (2 per night) within this range which gives little opportunity to apply a time restriction criterion.
Examination of nightly OH measurements show that tidal magnitudes are small (diurnal tide is <2K and semidiurnal <1K) and averaging over 6 months, and with the vertical averaging kernel of MLS at this altitude will average out tidal effects.

13. L246---252 and Fig. 4. This is a very nice and colorful figure but is it really essential or a short sentence would be enough?

It was considered that the histograms were useful to show the variability and skewness in temperature distributions between years as a function of the mean annual winter temperature. It would be difficult to describe the range of distributions displayed in a short sentence.

14. L284. The choice of this altitude range sounds somehow arbitrary. The OH layer is not centered at 85km. Please, justify. What happens if you use 85---90km?

The altitude range selected was not arbitrary, but as the altitude range of the smallest chi-sqr fit to the OH temperatures. See discussion under first main comment.

15. L290. This is the main problem I see using MLS data for this study. Why don't you use SABER data, even with a smaller time---coverage during the winter months? It could do a good job in your 60---day running means solar and linear trends. Garcia---Comas et al. (2014) showed that MLS biases at these altitudes could be large on the South Pole.

We include the equivalent curve for SABER data in Part 2 (Figure 1) of this work on the QQO (available in discussions as acp-2019-1097). SABER's 60 day yaw cycle limits comparable observations over Davis to two intervals (day-of-year 75–140 and 196–262) and days prior to 106 and after 259 are outside the OH winter averaging interval. Therefore only days 106-140 and 195-259 are comparable and a large part of the winter months is not sampled. As a consequence SABER winter averages do not fit the OH observations at Davis as well as MLS.

[Figure]

SABER data and results are discussed extensively in the discussion and used to in section 5.3 to assess trends in the change in the height of the OH layer.

16. L293. The choice of this pressure level is not justified. You could be getting a good agreement due to a bias that could be masking a wrong selection of altitudes. Indeed, it is well known that the altitude of the OH---layer is variable (as you even mention in L582--584). Please, discuss this point.

We do not match absolute temperature but the variance in winter mean temperature anomalies over 14 years. The 0.0046hPa pressure level is selected as it yields the smallest chi-square of the pressure levels. See discussion above under main comments and item 20 below.

17. L299. Yes, you show this very nicely in Fig. 5, my favorite figure of the paper.

Yes, trends are not uniform, and the seasonal variation should be considered when comparing trends between observers. This is why we attempt to present some insight into seasonal and spatial trend variability of the mesopause region using the MLS dataset.

18. L302. Indicate the figure where this is shown.

Examination of the QQO feature is separated and undertaken in Part 2 of this work. The paper became too unwieldy to include both the trends analysis and QQO investigation in the one manuscript. This is brief statement on the seasonal variability of the QQO with the following sentence indicating it is discussed in more detail in part 2. The figure is provided as Figure 1b in Part 2. (reproduced below)

[Figure]

Figure 1. (a) Davis OH winter mean residual (solar response removed) temperatures (black line, standard error in mean error bars, dashed linear fit) compared with Aura/MLS [AMJJAS] mean residual temperatures for 0.0046 hPa (green line, standard error-in-mean error bars, dashed linear fit) and TIMED/SABER (pink dotted line, standard error-in-mean error bars). Gray dotted line is a sinusoid fit (peak-peak amplitude 3.0 K period 4.2 years). (b) Detrended Davis OH winter mean temperatures [AMJJAS] (black line, long-term linear fit removed) compared to FMA, MJJ and ASO monthly averages (red, green and blue points mark warm, mid and cold years for composite studies). (c) A Mortlet wavelet transform (order 6) of the detrended Davis OH winter mean temperatures. Coloured sections are power significant above 90% level as per colour bar. The black line indicates the cone of influence; points outside have been influenced by the boundaries of the time series.

19. L304. Aura/MLS trends "at 0.00464 hPa"

Text added as suggested

20. L307. How did you derive that this is the OH equivalent pressure level and that it does not change with latitude? Does SABER, measuring OH emission and temperature vs z and pressure, show a significant change of equivalent pressure with latitude?

As discussed above at the variation of the [AMJJAS] anomaly had the smallest chi-sqr compared to the measured OH winter average temperature anomaly at Davis.

The Figure below shows a comparison of SABER VER (altitude of peak) and corresponding pressure value (mb) for the years 2002-2018 (day 106 – 259 of each year) at Davis Station. The OH peak occurs at pressures in the range 0.00255 hPa to 0.003 hPa ($2.55 \times 10^{-3}$ - $3.0 \times 10^{-3}$ mb). This value lies between two of the Aura MLS levels (0.00464 hPa and 0.00215 hPa) on which the averaging kernels are centered and is in excellent agreement with the Davis MLS 4.2 Temperature PRES plot (page 3 above) included in response to the first Main Comment above. An inverse relationship between altitude and pressure at the OH peak is clearly evident, and justifies the selection of a pressure level comparison for OH temperatures over an altitude level.

[Figure]

On the question of the altitude (or pressure) of OH peak as a function of latitude, the figure below shows the variation of the altitude of the OH peak as a function of latitude and day-of-year for the year 2005 from SABER data. The overall pattern shown here is repeated year after year with only minor changes in detail.

[Figure]

Based on the two figures above, the MLS averaging kernel centered on 0.00464 hPa would appear to be a good representative for the temperature of the OH layer.

21. L308. Monthly anomalies? 60---night running means? What are these?

These are AMJJAS (southern hemisphere winter months) averages as described on L309.

22. L316. What is the origin of these enhance bands?

To the best of our knowledge, this is the first time that these bands have been reported. This observation is discussed in the context of similar work (both observational and modelling) in section 5.4.

23. Section 5.1. I enjoyed reading the review but it could probably be shortened and better organized and your results should be put into the context you describe. Are they reasonable? Do they agree? Does the seasonality of your data agree with other results? Does it agree with the expected variations? I suggest extending the title of this section in order to include ozone.

Section 5.1 subtitle was modified to include ozone.

Section 5.1 does compare the present results with other reports in their context, e.g., in lines: 426-420, 459-460, 503-504, 510-511, 558-559 (line numbers refer to the original manuscript).

We have significantly shortened (from 114 to 84 lines) and reorganised the discussion in this section from line 427-487 – omitted the discussion of ACE-FTS CO2 rates of change and merged and modified the section from L436-477 to replace with the following  -

"In a recent summary of progress in trends in the upper atmosphere, Laštovička (2017) identified greenhouse gases, particularly $CO_2$ as the primary driver of long-term trends there.  The overall effect of greenhouse gases at mesospheric altitudes is radiative cooling.  The important secondary trend drivers in the mesosphere and lower thermosphere (MLT) are stratospheric ozone, water vapour concentration and atmospheric dynamics.  Temperature trends are predominantly negative, and recent progress in understanding the magnitude of the cooling have arisen from confirmation and quantification of the role of ozone.  Lübken et al. (2013) present the results of trend studies in the mesosphere in the period 1961-2009 from the Leibniz-Institute Middle Atmosphere (LIMA) chemistry-transport model which is driven with European Centre for Medium–Range Weather Forecasts (ECMWF) reanalysis below 40 km, and observed variations of $CO_2$ and $O_3$.  They find that $CO_2$ is the main driver of temperature change in the mesosphere, with $O_3$ contributing approximately one third to the trend.  Linear temperature trends were found to vary substantially depending on the time period chosen primarily due to the influence of the complicated temporal variation of ozone.

Figure 3 of (Lübken et al., 2013) show a monotonically increasing trend on CO2 compared with a much more complicated temporal ozone variation (essentially constant until 1980, a rapid decrease from 1980-1995, followed by an increase since then.  Trends in ozone vary as a function of both altitude and latitude, with positive trends dominating in the lower stratosphere and mesosphere.  Increases in water vapour concentration are considered a secondary but non-negligible effect particularly in the lower thermosphere (Akmaev et al. 2006).  The trend effect of dynamics was found to be very slightly negative in the mesosphere, but very small compared with the radiatively induced trends.  At the mesopause, the trend due to dynamics was positive and significantly larger ($\sim$1 K/decade).  These results were found to be in good agreement with observations from lidars, Stratospheric Sounding Units (SSU) (Randall et al., 2009) and radio reflection heights which have decreased by more than 1 km in the last 50 years due to shrinking in the stratosphere/lower mesosphere caused by cooling."

The paragraph from lines 478-486 had also been omitted.

24. L440. Include reference.
25. L449. Include reference.
The reference for both lines is Laštovička (2017), which appears in the opening line of the paragraph containing those lines.  The reference was included again at the end of the (modified) paragraph.

26. L465. This is already said, also mentioning the same reference.
Replaced with the modified paragraph as above (Item 23).

27. L478---486. It is not clear to me what this has to do with this work. Hervig et al. (2019) paper mainly deals with the paradox on the solar response of H2O. Perhaps mentioning only their result related to temperature makes more sense. By the way, do you actually see a change of solar response of temperature from 1995 to 2018?

The Hervig et al (2019) reference and following discussion has been omitted in the revision of section 5.1.

The period 1995-2018 spans only 2 solar cycles, assessing the response of the two cycles independently would not be constructive considering the uncertainties.

28. L488. Also Solomon et al. 2018

The work of Solomon et al. (2018) using WACCM-X is cited in lines 662-671.  The two studies are discussed separately, since Solomon et al. use constant low solar activity conditions in an attempt to disentangle temperature changes arising from anthropogenic effects from solar induced variations.

29. L501---502 I do not understand "global averaged temperature (..) as a function of latitude". How does their month---to---month variability compare with the seasonal variability you derive? Please, overplot on Fig. 5 and discuss.

Corrected ".. zonal average ..". Sentence now reads " Qian et al. (2019) provide zonal averaged temperature trend values as a function of altitude (50-110 km) and latitude for each month (their Fig. 3) some of which are statistically significant."

30. L507. Please, write altitude
The word altitude *is* present in the sentence.

31. L559. Write "from 1995 to 2018". If you remove your QQO, don't you see such breaks? From your measurements, it seems that your trend is not monotonic. Quantify "no obvious sign".

Added "from 1995-2018" to the end of the sentence.

We are not certain how you propose we remove the QQO variation. In order that it be removed the process generating the QQO needs to be understood. We devote significant effort to examine this QQO signal in more detail in Part 2 of this work but are unable at this stage to isolate the mechanism, therefore have no index to model the QQO.

If there was a trend break in the period 1995-2018, one might expect to see a significant change in the trend, and in the solar response, when extending the period of the study from 16 years to 24 years. Such a change is not observed (as you note in your opening paragraph "the trends derived here coincide with their previous results and those for the solar response are only slightly different". To quantify, the coefficients over
16 years (French and Klekociuk, 2011) were
4.30 ± 1.02 K/100sfu (95% confidence limits 2.2 K/100sfu < S < 6.4 K/100sfu)
-1.20 ± 0.51 K/decade (95% confidence limits -0.14 K/decade < L < -2.26)
and 24 years were
4.79 ± 1.02 K/100sfu (95% confidence limits 2.6 K/100sfu < S < 6.99 K/100sfu)
-1.18 ± 0.87 K/decade (95% confidence limits 0.71 K/decade < L < -3.06)
Neither coefficient has changed outside the uncertainty.

We are unsure how you deduce from our measurements that the "trend is not monotonic"?

32. L584. Please, also mention Liu et al. (2006).

Added Liu et al. (2006).

33. L592. Garcia---Comas et al. (2017) also estimated the trend and solar response of OH* altitude and temperature from SABER.

The following sentence has been added after line 580. "García-Comas et al. (2017) reported a slightly larger decrease of 40 m/decade in SABER OH volume emission rate weighted altitude at mid-latitudes which accompanied a 0.7%/decade increase in OH intensity and a 0.6K/decade decrease in OH equivalent temperature."

34. P610. What is the expected change in MLS temperatures due to a change of 0.02---0.04 km? This may lead to a bias in the comparison between DAVIS and MLS. A way to test this could be done with SABER data by comparing temperature trends at a fixed altitude to temperature trends at a OH VER weighted temperature.

Since the averaging kernel of MLS temperatures at the OH altitude is of the order of 15km a change of 20 to 40 metres over 16 years (determined from SABER VER altitude 2002-2018, shown in Fig 7) is negligible!.

35. L657---661 Why are these results so different? Could the difference be due to sampling?

WACCM-X is a model, Aura/MLS is measured data.  The overall long term trend in WACCM-X at 85km is -0.52 ± 0.64 K/decade (Fig 10 in Qian et al., 2019) although zonally averaged monthly trends are more variable (Fig 3 in Qian et al., 2019).

36. L720. According to a possible contribution of 30% by ozone, these values have (at least) a 30% error.

The clause in parenthesis in lines 721-722 acknowledges the fact that the trend quoted ignores possible contributions of stratospheric ozone to the trend.

37. L723---726. Could you provide the reason for this minimum in 2009?

The minimum of both the solar cycle and QQO cycle occur in 2009. (See Fig 3a.)

38. L747. Mention here also the results from other OH observations (those listed in Table 1).

We have re-written these sections with the request to shorten the discussion in section 5. Trend comparisons with other ground based observers listed in Table 1 is covered in section 4.4

39. Fig1b. Do all years contain the same days of measurements from doy 106 to 259? If not, there could be a sample bias.

With the exception of 1999 when 2 intervals D095-126 and 213-249 were used to scan the OH(8-3) band and 1996 missing D176-202 all other years only have more than 85% nights within the winter averaging window sampled. (ie 85% of the nights have a valid nightly average temperature with at least 10 measurements that pass selection criteria). A sample bias could be introduced in computing the anomalies if there was a significant departure from the climatological mean in those intervals.

| year | 1995 | 1996 | 1997 | 1998 | 1999 | 2000 |
|---|---|---|---|---|---|---|
| of 153 | 133 | 111 | 131 | 151 | 81 | 145 |
| % nights | 86.9% | 72.5% | 85.6% | 98.7% | 52.9% | 94.8% |

| year | 2001 | 2002 | 2003 | 2004 | 2005 | 2006 |
|---|---|---|---|---|---|---|
| of 153 | 148 | 149 | 138 | 133 | 130 | 139 |
| % nights | 96.7% | 97.4% | 90.2% | 86.9% | 85.0% | 90.8% |

| year | 2007 | 2008 | 2009 | 2010 | 2011 | 2012 |
|---|---|---|---|---|---|---|
| of 153 | 146 | 143 | 146 | 137 | 150 | 150 |
| % nights | 95.4% | 93.5% | 95.4% | 89.5% | 98.0% | 98.0% |

| year | 2013 | 2014 | 2015 | 2016 | 2017 | 2018 |
|---|---|---|---|---|---|---|
| of 153 | 146 | 149 | 142 | 152 | 150 | 142 |
| % nights | 95.4% | 97.4% | 92.8% | 99.3% | 98.0% | 92.8% |

The sliding window (seasonal variation in trend parameters) give some indication of the range of trends obtained by selecting different intervals compared to the winter mean interval.

We have also tested the effect on the derived coefficients by omitting individual years sequentially from the model fit computation. These show the range of coefficients if a data gap for the entire winter interval was missing. All coefficients derived from the omitted year computations remain within the uncertainty limits of the solar cycle and long-term trend coefficients when all years are included.

[Figure]

The following paragraph was added to section 4.1 to address this concern
"The stability of trend coefficients was tested for the presence of sampling gaps in the OH temperature record. With the exception of 1999 when 2 intervals D095-126 and 213-249 were used to scan the OH(8-3) band and 1996 missing D176-202 all other years only have more than 85% nights within the winter averaging window sampled. (ie 85% of the nights

have a valid nightly average temperature with at least 10 measurements that pass selection criteria). A sample bias could be introduced in computing the anomalies if there was a significant departure from the climatological mean in those intervals. The test examined the effect on the derived coefficients by omitting individual years sequentially from the model fit computation. These show the range of L and S coefficients if a data gap for the entire winter interval was missing in a particular year. All coefficients derived from the omitted year computations remained within the uncertainty limits of the solar cycle and long-term trend coefficients when all years were included. "

40. Figure 2. Time coverage changes with doy. What is the effect of DW1?

Over the winter averaging window (D106-259) the diurnal time coverage varies from 13:13 hrs (D106, 15-Apr) to 19:00 hrs (D177, 21-Jun) and 10:45 hrs (D259, 15-Sep). From the OH nightly observations we observe that the amplitude of the diurnal tide is <2K and semidiurnal tide <1 K.

We sample the same hours on the same days each year, and average those over days 106-259 each year to derive the trends.

41. Fig 5.a. the minimum in solar trend is during the month when downwelling is maximum. This migh be an indirect compensation of the cooling due to the direct dependence of downwelling (warming) and solar flux (COMPROBAR!!!)

We agree that the (August) minimum in the solar trend may indeed be the result of indirect compensation of cooling by the warming from maximum downwelling at that time of the year. However, we consider that removal of the climatological mean calculated over two full solar cycles is the best that we can do to eliminate the substantial part of the seasonal trend, leaving the anomaly values as shown in Figure 1(b). The solar cycle and long-term trends were calculated simultaneously using linear regression on the anomaly. A non-linear effect of the seasonal behaviour of the OH layer on the solar cycle response, e.g., through downwelling of atomic oxygen rich air, which could lead to increase production of CO cannot be ruled out, but is beyond the scope of the present work.

42. Fig 5b. Perhaps you might be sounding different altitudes? What is the seasonal change of the altitude of the OH layer? Did you look at SABER data? Also, this might be connected to O3 trend seasonality or CO2 trend seasonality.

The altitude of the OH layer peak appears to have a substantial seasonal response at DAVIS with an altitude minimum in mid-winter. However, we believe that removal of the climatological mean over two full solar cycles together with fitting solar cycle and long-term trends simultaneously would take account of this variation. Part 2 of this work examines the seasonal and long-term relationships between observed trends in temperature, $CO_2$, $O_3$, and CO.

43. Figure 5. Define grey boxes and blue dots.

These *are* defined in the figure caption.

44. Figure 6. What is time sampling for MLS? Are you removing tides? Trends strongly depend on sampling (Rezac et al. 2018)

These are 6 month anomaly averages (AMJJAS and ONDJFM) in each grid box, as described in the caption. They are the same averaging intervals each year. Tides are small and average out over 6-month means.

45. Fig 6. Please, overplot trend at Davis on the 1d plot. Also indicate CAP and LEO position on the maps.

Have modified figure 6 to indicate the positions of all ground-based observations in table 1 where long-term trend and solar cycle coefficients have been provided for comparison. Note that the 1d plots are a zonal mean, and as the map plots show there is considerable spatial variability in the trends derived from MLS data.

46. Fig 6. b. The blue/red bands at 70N in the NH winter months look like the trend and solar response related to stationary PWs.

We are grateful for this interesting suggestion.  In the interest of keeping the manuscript to a reasonable length, we decided to defer a detailed study of this point, and we are reluctant to speculate on it without supporting evidence.

47. Fig. 7. What is the realtionship between this plot and the temperature anomaly? Can it help to explain differences between DAVIS and MLS?

Section 5.3 describes the effect that a vertical shift in the altitude of the OH layer would have on the emission weighted temperature which is measured by a ground-based instrument like that at Davis.  The purpose of Fig. 7 is to show that SABER data does not indicate a significant change in the altitude of the OH layer during the period 2002-2018. Therefore we can eliminate change in the altitude of the OH layer as a cause of temperature change detected by the spectrometer at Davis.

This Figure does not address any differences between Davis and MLS.

48. Table 1. Discuss these results in the text, particularly mention them in section 5.

The majority of these results are discussed in the text in section 4.4   Section 5 has been substantially re-written.  See response to point 38 above.

49. Table 1. Include MLS results in this list

This table was constructed on the basis of ground-based measurements only (as described in the caption), with the result that Aura/MLS results are not included. The global results of MLS are provided in Figure 6.